# Evaluating the Utility of Sentinel-1 in a Data Assimilation System for Estimating Snow Depth in a Mountainous Basin

Bareera N. Mirza[1], Eric E. Small[2], Mark S. Raleigh[1]

[1]College of Earth, Ocean, and Atmospheric Sciences, Oregon State University, Corvallis, OR, USA

[2]Geological Sciences, University of Colorado, Boulder, CO, USA

*Correspondence to*: Bareera N. Mirza (mirzaba@oregonstate.edu)

**Abstract:**

Seasonal snow plays a critical role in hydrological and energy systems, yet its high spatial and temporal variability makes accurate characterization challenging. Historically, satellite remote sensing has had limited success in mapping snow depth and snow water equivalent (SWE), particularly in global mountain areas. This study evaluates the temporal and spatial accuracy of recently developed snow depth retrievals from the Sentinel-1 (S1) C-band spaceborne radar and their utility within a data assimilation (DA) system for characterizing mountain snowpack. The DA framework integrates the physics-based Flexible Snow Model (FSM2) with a Particle Batch Smoother (PBS) to produce daily snow depth maps at a 500-meter resolution using S1 snow depth data. The S1 data were evaluated from 2017 to 2021 in and near the East River Basin, Colorado, using daily data at 12 ground-based stations for temporal evaluation and four LiDAR snow depth surveys from the Airborne Snow Observatory (ASO) for spatial evaluation. The analysis revealed significant inconsistencies in temporal and spatial errors of S1 snow depth, with higher spatial errors. Errors increased with time, especially during ablation periods, with an average temporal RMSE of 0.40 m. In contrast, the spatial RMSE exceeded 0.7 m, and S1 had poor spatial agreement with ASO LiDAR ($R^2 < 0.3$). Experiments with DA window sizes showed minimal performance differences for full-season and early-season windows. Joint assimilation of S1 snow depth with MODIS Snow Disappearance Date (SDD) yielded similar temporal errors but degraded performance in space relative to assimilating S1 alone, while SDD assimilation alone performed best spatially. While S1 may perform better in other regions or snow conditions, our findings are consistent with findings from other error analyses across the Western U.S. and suggest S1 has limited potential to improve snow DA in the East River Basin, specifically. Future work should address retrieval biases, refine algorithms, and consider other snow datasets in the DA system to improve snow depth and SWE mapping in diverse snow environments globally.

# 1. Introduction

Seasonal snow is a natural freshwater reservoir for around 20% of the world's population (Barnett et al., 2005) and plays a key role in the global energy budget. Global mapping of snow characteristics like snow water equivalent (SWE) is a critical challenge, especially in steep and complex mountain terrain where SWE has high spatial and temporal variability (Clark et al., 2011; Kinar and Pomeroy, 2015; López-Moreno et al., 2011; Raleigh and Small, 2017). The snow community has been developing ground measurement, modeling, and remote sensing techniques through campaigns (e.g., NASA SnowEx) to advance our capabilities for estimating snow characteristics. Despite these efforts, there is still no globally available snow dataset in mountainous terrain at appropriate resolutions (~100s m or finer). Where point SWE measurements exist (e.g., stations), the high spatiotemporal variability makes it challenging to extrapolate SWE across global mountain environments (Cluzet et al., 2022; Dozier, 2011; Dozier et al., 2016; Elder et al., 1998; Grünewald et al., 2010; Herbert et al., 2024; Molotch and Bales, 2006). Improving our ability to estimate SWE in complex mountain regions requires continued advancements in global snow measurement and modeling techniques.

Optical satellite remote sensing has been used extensively in estimating snow properties. Spaceborne photogrammetry is effective for retrieving high-resolution snow depth (Deschamps-Berger et al., 2022; Marti et al., 2016; McGrath et al., 2019). However, this estimation is unavailable during cloudy periods and in dense forests. Spaceborne laser altimetry, such as ICESat and ICESat-2 has demonstrated some capability in mapping snow depth (Besso et al., 2024; Deschamps-Berger et al., 2023; Hu et al., 2022; Treichler and Kääb, 2017), but errors are high and spatial-temporal sampling is sparse. Optical remote sensing techniques for snow cover area (SCA) mapping, using sensors like Landsat, MODIS, and Sentinel-2, permit monitoring snow cover extent (e.g., Stillinger et al., 2023). However, accurate mapping is hindered by cloud cover or dense vegetation (Aalstad et al., 2020). While optical remote sensing cannot measure SWE or snow depth, the information on SCA depletion can guide model-based estimates of SWE (e.g., Margulis et al., 2019; Rittger et al., 2016).

Passive microwave remote sensing retrievals have traditionally been used to retrieve snow depth (Foster et al., 1996; Kelly et al., 2003) and SWE (Chang et al., 1987; Derksen et al., 2005) however,

its coarse spatial resolution (up to ~25km) cannot capture the variability of mountain snowpack,
and its accuracy is reduced in wet or deep snow conditions (over 1m; Luojus et al., 2021). In
contrast, active microwave remote sensing, such as Sentinel-1's (S1) C-band synthetic aperture
radar (SAR) backscatter data, has shown potential in mapping mountain snow depth at a scale of
0.5 km or finer (Lievens et al., 2019, 2022). S1 data have the potential to improve understanding
of mountain snow distributions and associated processes like orographic precipitation dynamics
and streamflow generation (Brangers et al., 2024; Girotto et al., 2024). Nevertheless, S1 snow
depth data has two fundamental challenges: (1) the data are not temporally continuous (i.e., gaps
due to satellite repeat or removed pixels flagged as wet snow), and (2) snow density is required to
convert snow depth to SWE (snow depth to SWE = density x snow depth). Two possible solutions
are: (1) machine learning (Broxton et al., 2024, 2019; Dunmire et al., 2024) or (2) data assimilation
(DA) frameworks (Smyth et al., 2022; Alonso-González et al., 2022). Machine learning has been
used to produce temporally continuous snow depth data utilizing the S1 snow depth data (Broxton
et al., 2024), but may require region-specific training due to geographic variations in snowpack
characteristics, making it less suitable for global mapping. In contrast, the assimilation of remotely
sensed snow depth can improve snowpack estimation (including estimates of snow density) and
provide temporally continuous data (Deschamps-Berger et al., 2022; Girotto et al., 2020; Largeron
et al., 2020; Margulis et al., 2015; Smyth et al., 2019, 2020) . However, prior S1 DA studies have
typically been limited to early-season assimilation, (Brangers et al., 2024; De Lannoy et al., 2024;
Girotto et al., 2024) and little to no work has yet explored the full-season potential of the DA
pathway for SWE mapping using S1 snow depth.
While S1 snow depth data have the potential to guide DA systems to produce comprehensive
spatiotemporal mountain SWE maps, there have been discrepancies in the reported spatial and
temporal errors, which prompt questions about its reliability. Independent evaluations have
reported large errors (Broxton et al., 2024; Dunmire et al., 2024; Hoppinen et al., 2024; Sourp et
al., 2025) relative to LiDAR snow depth from Airborne Snow Observatory (ASO; Painter et al.,
2016) surveys and gridded snow datasets (Broxton et al., 2024). Sourp et al. (2025) found that S1
retrievals exhibit no clear error pattern but consistently underestimate snow depth, particularly
before and after the melt period. Broxton et al. (2024) found mean bias from 0.27m to 0.25m when
compared against ASO LiDAR flights, depending on the removal of "flagged wet pixels".
Hoppinen et al. (2024) found that these flagged pixels do not fully eliminate wet pixels and remove
some shallow dry snow pixels, resulting in a loss of usable data. Hoppinen et al. (2024) showed
little to no spatial correlation between S1 and LiDAR data, contrary to R ~0.52 reported by Lievens
et al. (2019) over the Northern Hemisphere. Evaluation studies (Broxton et al., 2024; Hoppinen et
al., 2024) validate S1 data on a handful of dates near/after peak SWE with LiDAR surveys in the
western U.S., while Lievens et al. (2022) conducted validation against point-scale time series at
1000s of locations in the European Alps and other global mountain ranges. Previous studies have
shown that C-band S1 is ineffective near or after peak SWE due to the high liquid water content in
the snowpack, which leads to the attenuation and absorption of microwave energy (Gagliano et al.,
2023; Nagler et al., 2016), leading to increased uncertainties in the retrieved snow depth. Thus, the
spatial evaluation data (airborne LiDAR) are mostly available later in the snow season when S1
data are less reliable. Gascoin et al. (2024) highlights this limitation and recommends two strategies
using S1 snow depth in a DA system: (1) assimilating S1 only during the early season when snow
is dry, and (2) implementing a joint assimilation of S1 snow depth along with other remote sensing
data (e.g., the snow disappearance date from optical snow-covered area data). To our knowledge,
no studies have tested these strategies with S1 snow depth.
This paper aims to understand the spatiotemporal error discrepancies in S1 snow depth and assess
how these data can be used in a DA system, both alone and in combination with other remote
sensing data. First, we evaluate the S1 snow depth data (500 m resolution) using ground-based data
at 12 sites and four airborne LiDAR flights in the well-studied East River Basin (Colorado, USA).
Second, we develop a DA system using a physics-based model leveraging the recent development
of the open-source toolbox Multiple Snow Data Assimilation System (MuSA) (Alonso-González
et al., 2022) and deploy our DA system in a cloud-based environment. Using this system, we test
whether assimilation of S1 snow depth across the full season (all observations) versus early season
(when it is assumed to be most reliable) improves snow depth estimation in time and space, relative
to snow simulations without assimilation. Finally, we test the joint assimilation of S1 snow depth
and Moderate Resolution Imaging Spectroradiometer (MODIS) snow disappearance date (SDD)
(Crumley et al., 2020) following the recommendation of Gascoin et al. (2024). We address three
research questions:

1. How do Sentinel-1 snow depth errors vary across space and time?

2. What is the relative value of using full-season Sentinel-1 data compared to early-season data in a data assimilation framework?

3. Can the joint assimilation of Sentinel-1 snow depth and MODIS snow disappearance date enhance DA accuracy?

# 2. Study Location and Data:

## 2.1. Study Location:

The study area is the East River Basin, Colorado (ERB Figure 1a), which has a ~748 km² area and features alpine tundra at higher elevations, montane forest at middle elevations, and prairie snow climates in lower elevation valleys (Sturm and Liston, 2021). The ERB has an average elevation of 3266m with 1,420 m topographic relief. It is characterized by cold winters, with a mean annual temperature of 0°C and mean annual precipitation of 1200 mm, mostly as snow (Daly et al., 1994; Hubbard et al., 2018). This cold, dry climate, along with moderate canopy cover, makes the ERB a representative testbed for assessing S1, which tends to struggle in complex terrain (e.g., wet snow, dense forest; Lievens et al., 2019). Furthermore, it has been well-studied and has extensive validation data through multiple scientific field campaigns, such as NASA SnowEx, ASO, the DOE SAIL (Feldman et al., 2023), and NOAA SPLASH (De Boer et al., 2023). It also includes long-term NRCS SNOTEL sites within the basin (Butte) and nearby (e.g., Schofield Pass). In addition, ground measurements are available through a research site established in late 2018 on Snodgrass Mountain (Bonner et al., 2022), which was included in the 2020 NASA SnowEx campaign. Additionally, the ERB has been mapped by airborne LiDAR flight surveys carried out by ASO (Painter et al., 2016), which provided 50 m snow depth measurements twice in two study years (2018 and 2019). The study period spans water years 2018-2021, coinciding with the availability of S1 data and ASO data.

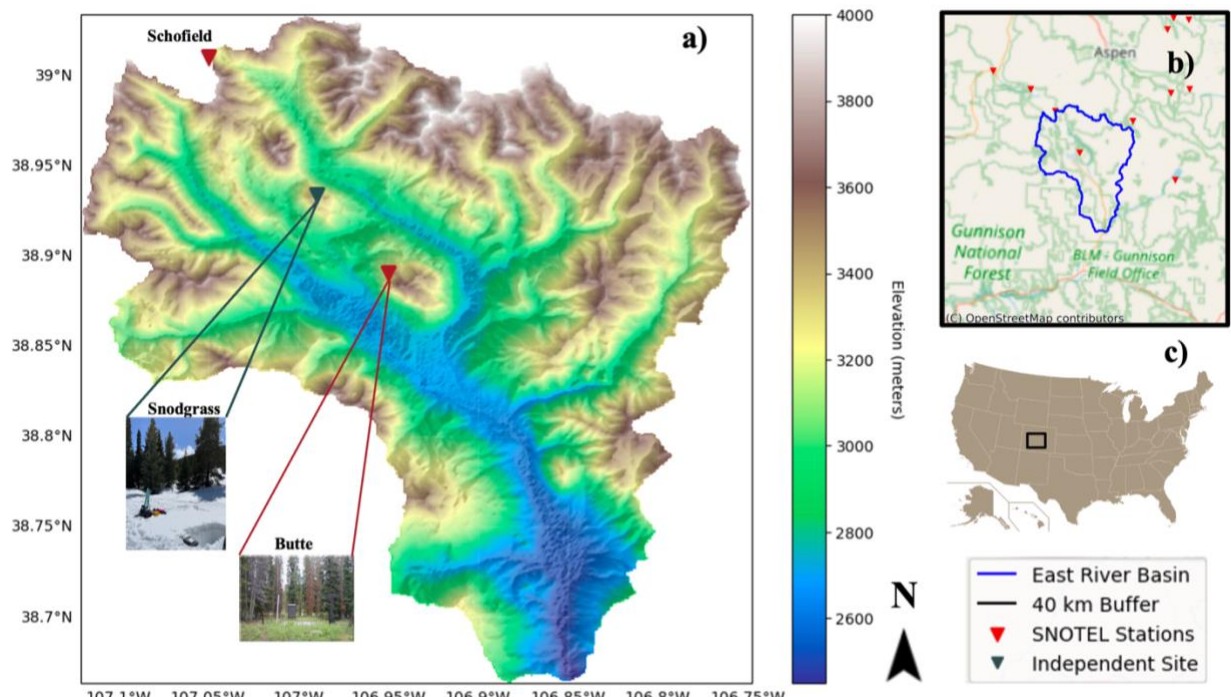

**Figure 1: (a)** Elevation map of the East River Basin (Jarvis et al., 2008) with Snodgrass and SNOTEL (USDA,NRCS, 2024) stations within and near the basin. **(b)** SNOTEL stations within 40km from the boundary of the basin. **(c)** Location of the East River Basin shown relative to the Western United States.

## 2.2. Data

### 2.2.1. Meteorological forcing:

ECMWF ERA5-Land reanalysis data (Copernicus Climate Change Service, 2019; Hersbach et al., 2020) are used as the source of meteorological forcing data as they are available globally at a 0.1-degree (9 km) resolution hourly from 1950 to the present. The data, accessed from Google Cloud Storage, include incoming shortwave and longwave radiation [W m⁻²], total precipitation (liquid and solid) [kg m⁻² s⁻¹], surface atmospheric pressure [Pa], 2 m air temperature [K], 2 m relative humidity [%], and 10 m wind speed [m s⁻¹]. The data were not downscaled to the S1 500 m resolution because the DA approach implicitly accounts for fine-scale variability during assimilation (see below). In this implementation, a model ensemble is first generated at the original 9 km resolution and then regridded by nearest neighbor to 0.5 km resolution before applying the Particle Batch Smoother (PBS), effectively capturing the downscaling process (Bachand et al., 2025; Girotto et al., 2024; Smyth et al., 2019, 2020).

### 2.2.2. Sentinel-1 Snow Depth:

We analyzed the 500 m S1 snow depth product (Lievens et al., 2022), which is available across
mountainous regions of the Northern Hemisphere from 2016 to 2021. This dataset is derived from
C-Band (5.4 GHz) SAR backscatter measurements collected by the ESA Sentinel-1A and Sentinel-
1B satellites. The snow depth retrieval algorithm utilizes temporal variations in the ratio between
cross-polarized and co-polarized backscatter signals, which are sensitive to snow depth in dry
snow. Snow depth retrievals have reported mean absolute errors of 0.18m over the Northern
Hemisphere (Lievens et al., 2019). However, the S1 performance degrades in challenging
conditions (snow that is wet, shallow, patchy, or in forests) with a mean absolute error of about
1.3m at a depth of 3.5m and above (Lievens et al., 2022). Evaluation and development of this
dataset is ongoing. The latest 500m dataset includes a quality flag for wet snow conditions, which
has potential use for identifying and removing pixels with higher retrieval errors, particularly
during the ablation season. Wet snow leads to the absorption and attenuation of radar signals, which
makes snow depth mapping unreliable. The frequency of observations varies by region depending
on the satellite overpass schedule. However, the dataset offers temporal resolution primarily on a
daily to weekly basis, with most intervals being less than one week over the ERB.
2.2.3. Snow Disappearance Date (SDD):
We conduct multiple DA experiments (see Section 3.4), some of which utilize SDD as an input in
the assimilation to test whether S1 provides new information. We utilize SDD rather than daily
time series of snow-covered area for simplicity, and recognizing that SDD has been found to have
high correlation with maximum SWE (Trujillo and Molotch, 2014). We also include SDD in a
joint assimilation experiment to inform our model about end-of-season snowpack conditions (i.e.,
Gascoin et al., 2024), especially since S1 is only available through April 30[th] and has limitations
during the melt season due to signal attenuation (Lievens et al., 2022). We derive SDD from
different sources depending on the experiment. For the temporal evaluations, we derived SDD
from SNOTEL by finding the first zero snow depth value after peak snow depth. For spatial
evaluations, we derived SDD from MODIS snow cover data using annual summary metrics based
on the daily Normalized Difference Snow Index (NDSI) from the MOD10A1 product with a
threshold of 0.15, leveraging the SnowCloudMetrics algorithm implemented on Google Earth
Engine (Crumley et al., 2020). The MODIS SDD is determined by identifying the last five snow-
free days (backward approach) preceded by at least five consecutive snow-covered days. This
approach minimizes the impact of transient late-season snow events while ensuring a consistent
detection of snow disappearance.
## 2.2.4. Evaluation Data:
We conduct temporal and spatial evaluations of snow depth, both prior to S1 assimilation and
posterior to assimilation. Temporal evaluation utilizes daily snow depth from 11 NRCS SNOTEL
stations within 40 km of the basin. Since the S1 change detection algorithm was optimized using
SNOTEL data (Lievens et al., 2019), an independent research site at Snodgrass Mountain (Figure
1) was also utilized to assess the reliability of S1 snow depth time series away from SNOTEL sites.
The Snodgrass Mountain site provides snow depth data, along with numerous monthly in situ snow
pit measurements from February to May. These snow pits, located in both open and forested areas
with a spatial extent of 1 km, are ideal for evaluating the 500 m S1 grid. S1 and posterior snow
depths are evaluated against observations from these sites for WYs 2018–2021 for SNOTEL sites
and 2019–2021 at the Snodgrass site.
The spatial evaluation uses four ASO LiDAR flights, with 50 m snow depth data collected near
peak SWE (i.e., late March / early April) and in the mid-melt season (i.e., May or June) in 2018
and 2019 over the ERB. The data were resampled to match the 500 m S1 grid resolution using
bilinear interpolation.
# 3. Methods:
## 3.1. DA Methodology
DA integrates model estimates with observations to estimate the most representative state of a
system (e.g., SWE) with uncertainty. DA can account for observational uncertainty (e.g., S1 snow
depth, MODIS SDD), and some DA approaches can provide physically consistent estimates of
multiple snow states when implemented with a physically-based snow model. In this study, we
develop and deploy a Python-based DA system in the Google Cloud Platform. Within the cloud,
the system accesses ERA-5 Land meteorological data and uses it as input forcing into the MuSA
toolbox, which communicates with the physically-based Flexible Snow Model version 2 (FSM2)
model to generate ensembles of model simulations at a 9-km spatial resolution. The snow model
ensembles from MuSA are combined with snow observations (e.g., S1 snow depth and/or SDD)
via the particle batch smoother (PBS). Through assimilation of snow data with a wide ensemble
(section 3.2), PBS also serves as a downscaling technique(Bachand et al., 2025) from the 9-km
ERA5-Land forcings to the 500 m grid. The flow chart in Figure 2 and the sections below detail
the key components.

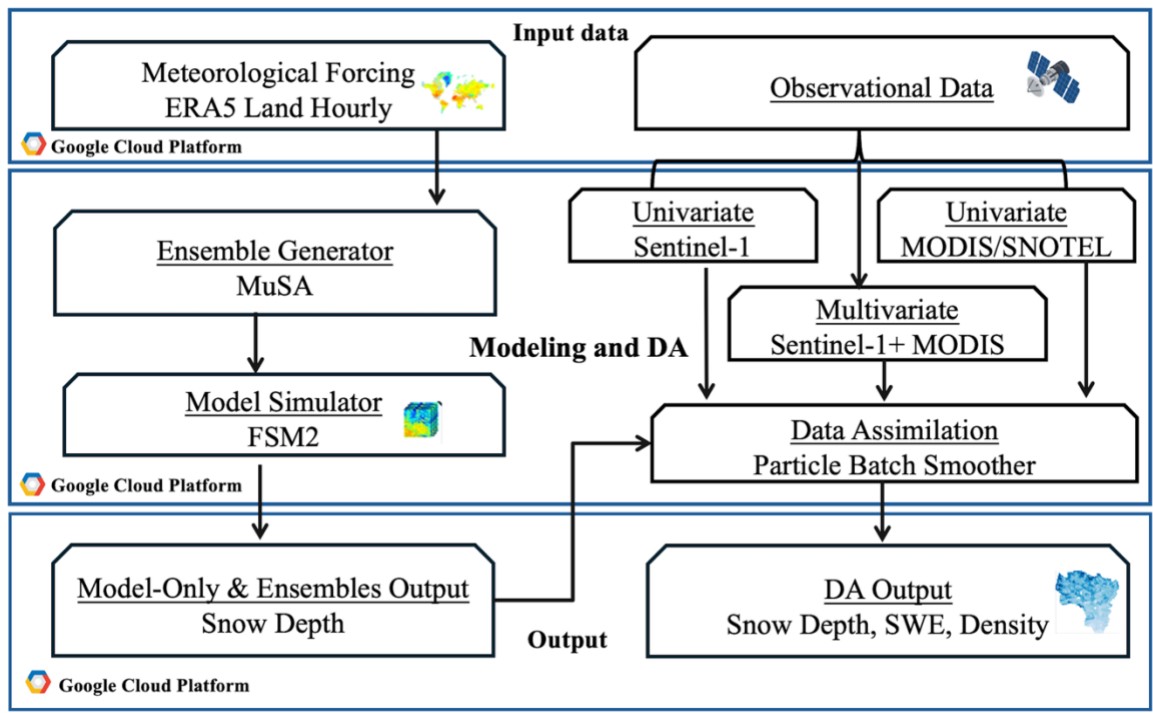


**Figure 2**. Flowchart illustrating the input, model, and DA framework, and outputs (top to bottom).

## 3.2. Snow Model and Ensemble Generation

FSM2 (Essery, 2015; Essery et al., 2024) is a multi-physics model that simulates the mass and
energy balances of snow on the ground (including under forest canopies), employing three snow
layers by default. The model incorporates conservation equations for liquid water, ice, and internal
energy, offering a detailed representation of snowpack processes. FSM2's flexibility permits
independent activation or deactivation of parameterizations, facilitating a range of model
configurations (Essery, 2015; Essery et al., 2024). We utilize the most complex configuration of
FSM2 to simulate internal snowpack processes, which we now briefly summarize. Albedo is
calculated based on snow age, decreasing its value with time, and increasing it with fresh
snowfalls. Snowpack thermal conductivity is determined by snow density, which is computed
based on overburden and thermal metamorphism. Turbulent energy fluxes are computed based on
bulk aerodynamic theory, and Monin-Obukhov adjustment for atmospheric stability is activated as

part of the calculation process. Meltwater percolation in the snowpack is computed using gravitational drainage. The selected configuration has demonstrated success in related DA studies (Smyth et al., 2022).

MuSA is an open-source, ensemble-based snow DA tool designed to assimilate multiple observations with FSM2 while considering various sources of forcing and measurement uncertainty (Alonso-González et al., 2022). We use MuSA to generate ensembles of FSM2 at a point scale. We perturbed the 9-km ERA5-Land hourly meteorological forcing data by drawing spatially independent, random perturbation parameters from a logitnormal distribution for precipitation (to avoid negative values) and from a normal distribution for temperature and LW radiation (Table 1). For precipitation, we applied bounds of [0, 4] with $\mu = 0$ and $\sigma = 1$, capturing a wide range of basin-scale variability. For temperature and LW, we applied $\mu = 0$ and $\sigma = 2.5$ and $\mu = 0$ and $\sigma = 20.8$, respectively. Several combinations of bounds, standard deviation, and mean values were tested, and the final configuration was selected to be broader than typical ranges to account for the fact that ERA5-Land was not downscaled prior to assimilation and to better represent spatial variability in the East River Basin, Colorado. We tested and visualized the resulting spread to confirm realistic precipitation, temperature, and LW scenarios, and found that more than 98% of observed S1 snow depth values fell within the ensemble bounds during assimilation. Overall, this perturbation strategy was chosen to ensure the ensemble spans a wide range of meteorological conditions and captures a realistic range of possible snow depths.

**Table 1.** Generation of perturbed inputs for each particle member.

| Variable | Unit | Adjustment | Distribution | Lower Bound | Upper Bound | Std. dev. | Mean |
|---|---|---|---|---|---|---|---|
| **Precipitation** | mm/h | Multiplicative | Logit-normal | 0 | 4 | 1 | 0 |
| **Longwave radiation** | W/m$^2$ | Additive | Normal | - | - | 20.8 | 0 |
| **Temperature** | °C | Additive | Normal | - | - | 2.5 | 0 |

Once the forcing data are perturbed, MuSA runs FSM2 to generate an ensemble of distinct snow simulations. Similar to Bachand et al. (2025), we used 100 ensemble members (particles) to ensure computational efficiency while adequately capturing the variability in the prior distribution. For

each particle, we record the SDD (i.e., the first snow-free date after peak snow depth), for use in multiple DA experiments (see Section 3.4).

## 3.3. Particle Batch Smoother – DA Algorithm

Once the ensembles are generated, we assimilate snow observations (e.g., snow depth and/or SDD) using the PBS algorithm. Separately, we tested other DA approaches, namely the Particle filter (PF), both with constant and dynamic errors, the latter of which increases errors in time as revealed in our temporal analysis as SNOTEL sites (Figure 4). Despite having constant observational error, PBS had lower RMSE than both PF implementations (Supplement Figure S1, Table S1), particularly during the ablation seasons, likely due to its smoothing properties. We therefore selected the PBS with temporally constant snow depth error for subsequent analysis. PBS is well-suited as a downscaling tool as well (Bachand et al., 2025), further motivating its selection in this study.

PBS employs a Bayesian approach, representing state variables like snow depth and SDD through a collection of particles, where each particle represents a possible system state. For observations within the assimilation window (e.g., snow season), PBS updates particle weights based on their likelihood of representing the true state, using the likelihood function from Margulis et al. (2015). This process combines the prior probability density function (PDF) with the likelihood to estimate the posterior PDF of snowpack variables, such as snow depth and SWE:

$$p_{Z|Y}(Z|Y)=p_V(Z-M_j^-)=\frac{1}{\sqrt{(2\pi)^{N_{obs}}|C_V|}}\times\exp\left[-0.5\left(Z-M_j^-\right)^{\top}C_V^{-1}\left(Z-M_j^-\right)\right] \qquad \text{Equation (1)}$$

Where $p_{(Z|Y(Z|Y))}$ is the likelihood function, which represents the probability of observing the measurement $p_V(V)$ is the probability density function (PDF) of the measurement error vector V, $Z$ is the observed measurement (e.g., S1 snow depth, MODIS SDD), $M_j$ is the modeled snow variable (e.g., snow depth or SDD), and $C_v$ is the error covariance matrix of the measurement error vector V, which represents the uncertainty in the measurement.

## 3.4. DA Experiments

288 To address Research Questions 2 and 3, we conducted four experiments, building on the findings
289 of Research Question 1 (S1 error analysis). The DA experiments are as follows (Figure 3):

1. Snow Depth (**Hs**) **F**ull Window **(Hs-F):** Assimilation of all Hs, including dry and wet snow pixels, from November 15th to April 30th. S1 has data only until April 30th, and 83% of S1 data points before November 15th have zero or less than 10cm snow depth values. Thus, we call Hs-F a full-window experiment.

2. Snow Depth (**Hs**) **E**arly Window **(Hs-E):** Assimilation of early-season (November 15$^{th}$ to January 15$^{th}$) Hs, where the early window has a lower observational error than the full window (based on error analysis, section 4.1).

3. Joint Assimilation of SDD and Snow Depth in the **F**ull Window **(SDD+Hs-F):** Combined assimilation of SDD and Hs over the same window as Hs-F. SDD data were derived from SNOTEL for temporal evaluation and MODIS for spatial evaluation.

4. **S**now **D**isappearance **D**ate Only **(SDD):** Assimilation of SDD data alone, derived from SNOTEL (temporal evaluations) or MODIS (spatial evaluations). This serves as a baseline or control experiment to understand whether S1 snow depth adds new information.

SDD is the baseline experiment, and each of the experiments are evaluated against 12 ground-
based stations temporally and 4 ASO LiDAR flights spatially (Section 2.2.2). In experiments Hs-
F, Hs-E, and SDD+Hs-F, the observational uncertainty value (Equation 1) for S1 snow depth is the
RMSE when evaluated against SNOTEL for all four years. The SDD observational uncertainty is
set to five days, consistent with the retrieval algorithm in SnowCloudMetrics (Crumley et al., 2020;
Slater et al., 2013). In the joint assimilation (SDD+Hs-F), both datasets are weighted equally to
balance their respective influence in DA. The joint likelihood is computed in log space under the
assumption of independence between the S1 snow depth and SDD observations. This equal-weight
approach for joint assimilation is necessary for two reasons: (1) there is an imbalance in the number
of observations for Hs (dozens of observations) versus SDD (1 value) for each location and year,
and (2) the Hs time series exhibits temporal autocorrelation.

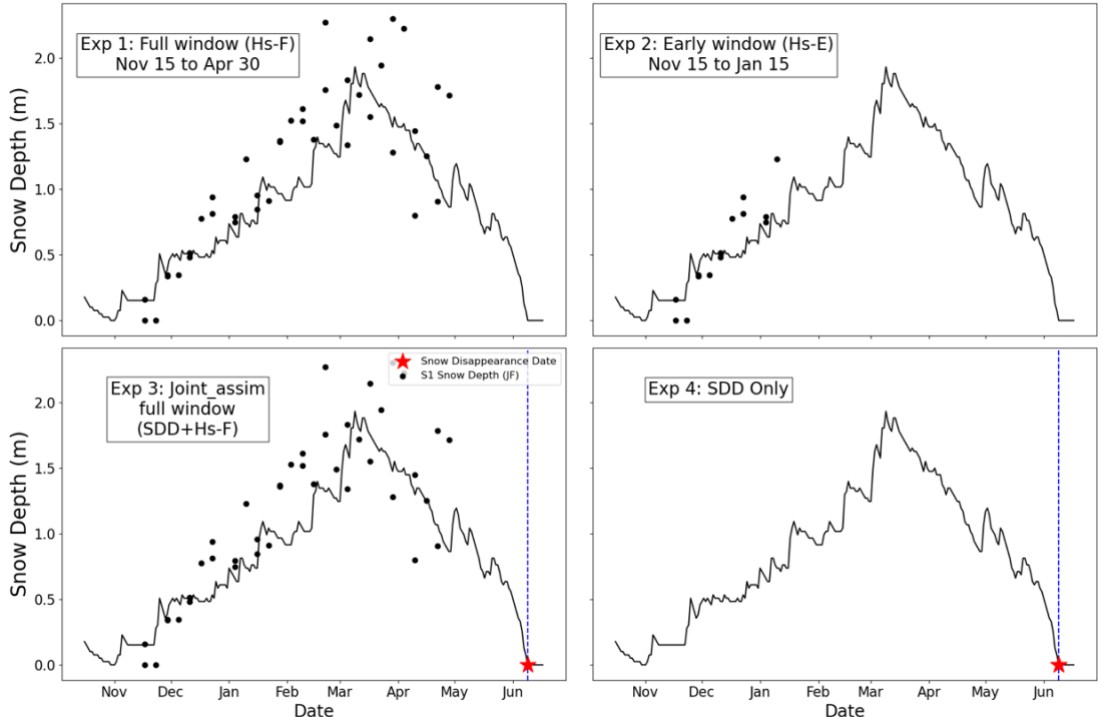


**Figure 3**. Conceptual illustration of four experiments. Black dots indicate S1 observation data, black lines represent posterior estimates, and red stars (with blue dashed line) denote the SDD.

## 3.5. Evaluation

We evaluated snow depth (i.e., S1 observations and PBS experiments) with standard valuation metrics, including coefficient of determination ($R^2$), root mean squared error (RMSE), mean bias, mean absolute error (MAE), and relative mean absolute difference (RMAD). These were selected to enable comparisons to other studies (Lievens et al., 2019; Broxton et al., 2024; Hoppinen et al., 2024). To avoid seasonal inflation of $R^2$ in time series, where snowpack seasonality alone can yield artificially high correlations, we restrict temporal validation to dates where both modeled and observed snow depth are present (i.e., excluding summer/no-snow periods). These dates span the snow season (October 1 to April 30 or until disappearance in the SDD case), ensuring metrics evaluates on aligned timesteps. For spatial validation (e.g., S1 vs. LiDAR), metrics are computed for individual dates to avoid temporal autocorrelation effects. While $R^2$ is sensitive to underlying model forcings, we use it alongside RMSE, MAE, and mean bias to provide a comprehensive performance evaluation. We also evaluated MAE and RMAD, which are defined as:

$$MAE = \frac{1}{n}\sum|y_i - \hat{y}_i| \qquad \text{Equation (2)}$$
$$RMAD = \frac{\frac{1}{n}\Sigma_i |\hat{y}_i - y_i|}{\overline{y}}$$        Equation (3)
where $y_i$ is the observed value (e.g., LiDAR, SNOTEL, field data), $\hat{y}_i$ is the predicted value (e.g.,
PBS posterior mean), $\overline{y}$ is the mean of all observed/validation data values, and $n$ represents the
total number of data points.
We also evaluated the probabilistic performance of the posterior ensemble simulations using the
Continuous Ranked Probability Score (CRPS) at 12 point-scale observation sites. CRPS is a
proper scoring rule that accounts for both the reliability and sharpness of probabilistic estimates
from ensemble simulations, with lower values indicating better performance.  We use the discrete
form of CRPS as described by (Hersbach, 2000), which is equivalent to the continuous CDF-
based definition (Matheson and Winkler, 1976) when applied to finite ensemble samples.:
$\text{CRPS}(y_t, \{\hat{y}_{t,i}\}) = (1/N) \Sigma_{i=1}^{N} |\hat{y}_{t,i} - y_t| - (1/(2N^2)) \Sigma_{i=1}^{N} \Sigma_{j=1}^{N} |\hat{y}_{t,i} - \hat{y}_{t,j}|$    Equation (4)
Where $y_t$ is the observed snow depth (e.g., from SNOTEL) at time t, $\hat{y}_{t,i}$ is the snow depth
predicted by the i-th ensemble member at time t, and N is the total number of ensemble
members.
To quantify the added value of assimilation relative to the reference (open-loop) simulation, we
computed the CRPS Skill Score (CRPSS) following the symmetric formulation of (Cluzet et al.,

347    2022)

348        $CRPSS(E,R) = \{ 1 - CRPS(E)/CRPS(R),$      $if\ CRPS(E) < CRPS(R);$

349        $CRPS(R)/CRPS(E) - 1, otherwise \}$          Equation (5)

This formulation bounds CRPSS between [–1, 1], making it possible to directly compare and
average positive (improvement) and negative (degradation) values.

# 4. Results

## 4.1    Error Analysis

Errors in the S1 500m snow depth data were evaluated temporally at 11 SNOTEL stations and an
independent site (Snodgrass) from 2018–2021 and spatially relative to ASO LiDAR flights in 2018
and 2019. We included wet/flagged pixels in all our analyses/experiments because the removal of
flagged pixels resulted in performance degradation (results not shown). The results are shown in
Table 2.
**Table 2.** Average $R^2$, RMSE, Mean Bias, MAE, and RMAD values of **(a)** S1 against 11 SNOTEL sites and the
Snodgrass site. (b) S1 against ASO 50m LiDAR snow depth aggregated to 500m.

| | **(a)** Temporal evaluation – average across all stations (m) | | | | | **(b)** Spatial evaluation – S1 against ASO LiDAR (m) | | |
|---|---|---|---|---|---|---|---|---|
| **Year** | **2018** | **2019** | **2020** | **2021** | **Average** | **2018** | **2019** | **Average** |
| **R²** | 0.74 | 0.80 | 0.72 | 0.72 | 0.74 | 0.19 | 0.27 | 0.23 |
| **RMSE** | 0.32 | 0.39 | 0.39 | 0.53 | 0.40 | 0.74 | 0.91 | 0.82 |
| **Mean Bias** | 0.08 | -0.07 | 0.09 | 0.26 | 0.09 | -0.47 | -0.26 | -0.36 |
| **MAE** | 0.21 | 0.26 | 0.27 | 0.38 | 0.28 | 0.60 | 0.72 | 0.66 |
| **RMAD** | 0.61 | 0.39 | 0.47 | 0.68 | 0.53 | 0.77 | 0.43 | 0.60 |

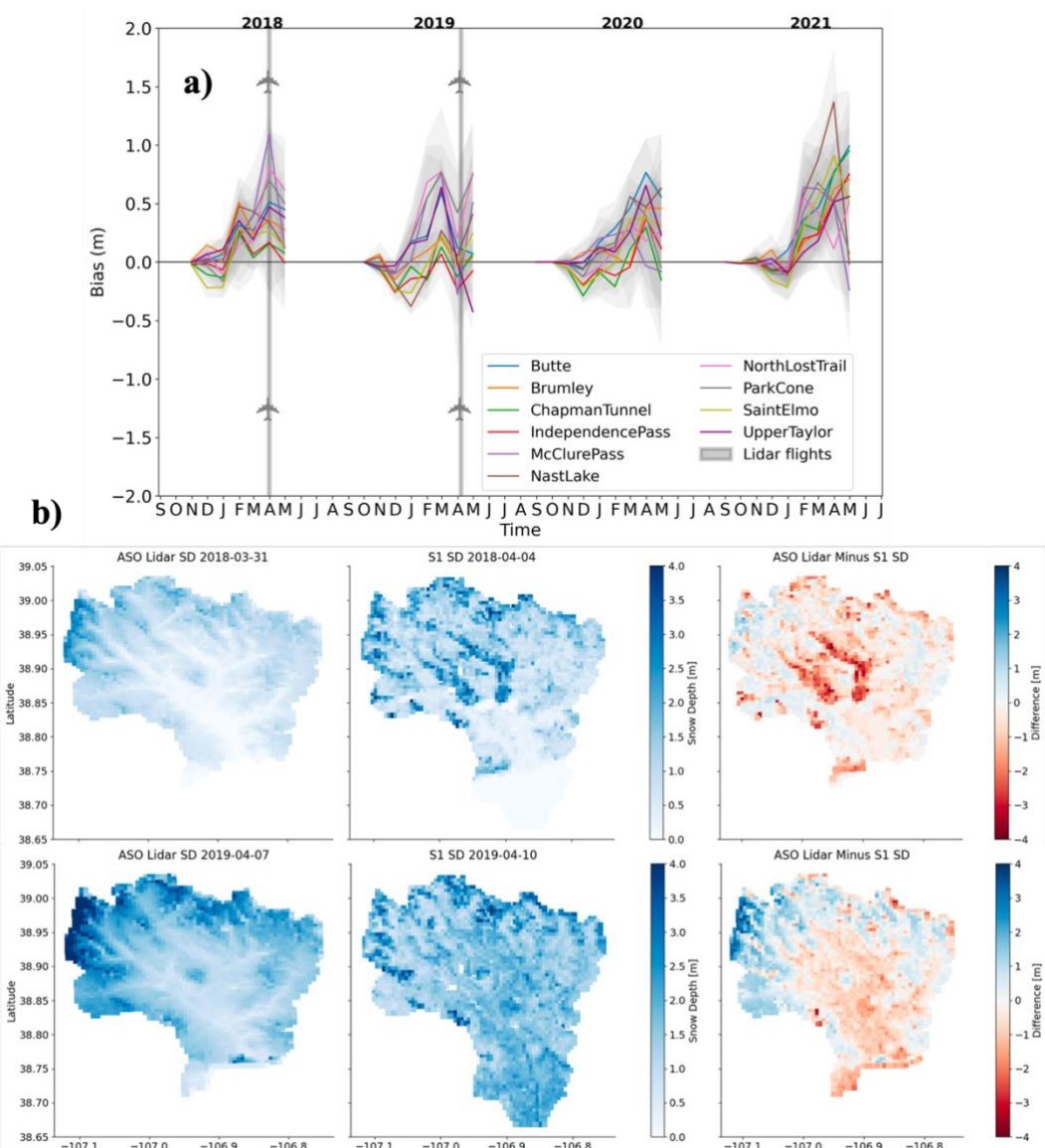

**Figure 4:** (a) Time-varying errors at the 11 SNOTEL sites with the timing of LiDAR flights marked with thick grey lines. The solid lines represent the monthly mean bias, while the shaded region represents the ± one standard deviation from the from daily bias values, indicating intra-month variability. Each station is represented by a different color. (b) Comparison of ASO LiDAR and Sentinel-1–derived snow depth estimates on temporally proximate dates.

When evaluated temporally against stations, S1 had an average bias of +0.09 m, RMSE of 0.40 m and $R^2$ of 0.74. In contrast, when evaluated spatially against ASO, S1 had higher bias magnitude ( -0.36 m), a higher RMSE (0.82 m), and a lower $R^2$ (0.23) (Table 2). These results indicated that S1 performed better in capturing temporal variations compared to spatial patterns. S1 errors tended to

increase over time across all stations (Figure 4a), with the lowest errors in the early season and the highest errors (~ 0.4m) near peak snow depth. The timing of the two LiDAR flights (Fig. 4b) coincided with the time of year when errors tend to be high (thick grey vertical lines, Fig. 4a).

## 4.2. Temporal Experiments:

Before analyzing the temporal performance of our assimilation windows, we first evaluated the overall skill of the posterior ensembles. We computed the CRPS (Eq. 4) across all sites and water years, which directly measures the distance between ensemble predictions and observations. The mean CRPS was 0.21 m, showing that posterior ensembles were, on average, within ~21 cm of observed snow depth and maintained skill across a wide range of snowpack conditions (Figure S2). CRPSS was then used to compare each experiment against the open-loop reference. As an uncertainty-aware metric, positive CRPSS values indicate improvement relative to the reference. Systematics CRPSS averages ranged from 0.08 to 0.22 (Table S2), reflecting modest but consistent gains from assimilation. Higher values for SDD and joint assimilation highlight the added value of snow disappearance information, while lower values for Hs-only experiments underscore the noisier character of Sentinel-1 snow depth.

With this confidence in ensemble performance, we assessed the performance of S1 assimilation with respect to window size, i.e., full window data assimilation (Hs-F) compared to the early window (Hs-E) when errors were lower (Fig. 4a). The results are shown in Figure 5 for two sites: (1) an independent site (Snodgrass), which was not used in the optimization of the S1 dataset, and (2) a representative SNOTEL site. The time series of the two sites shows that the performance of Hs-F and Hs-E is not consistent across the years. Hs-F (purple line) performed marginally better than Hs-E (green line) in WY 2019 and similarly in 2020 and 2021 for the Snodgrass site. Likewise, at Chapman Tunnel, Hs-F performed better than Hs-E in WY 2018, poorly in 2021, and similarly to HS-E in 2019 and 2020.

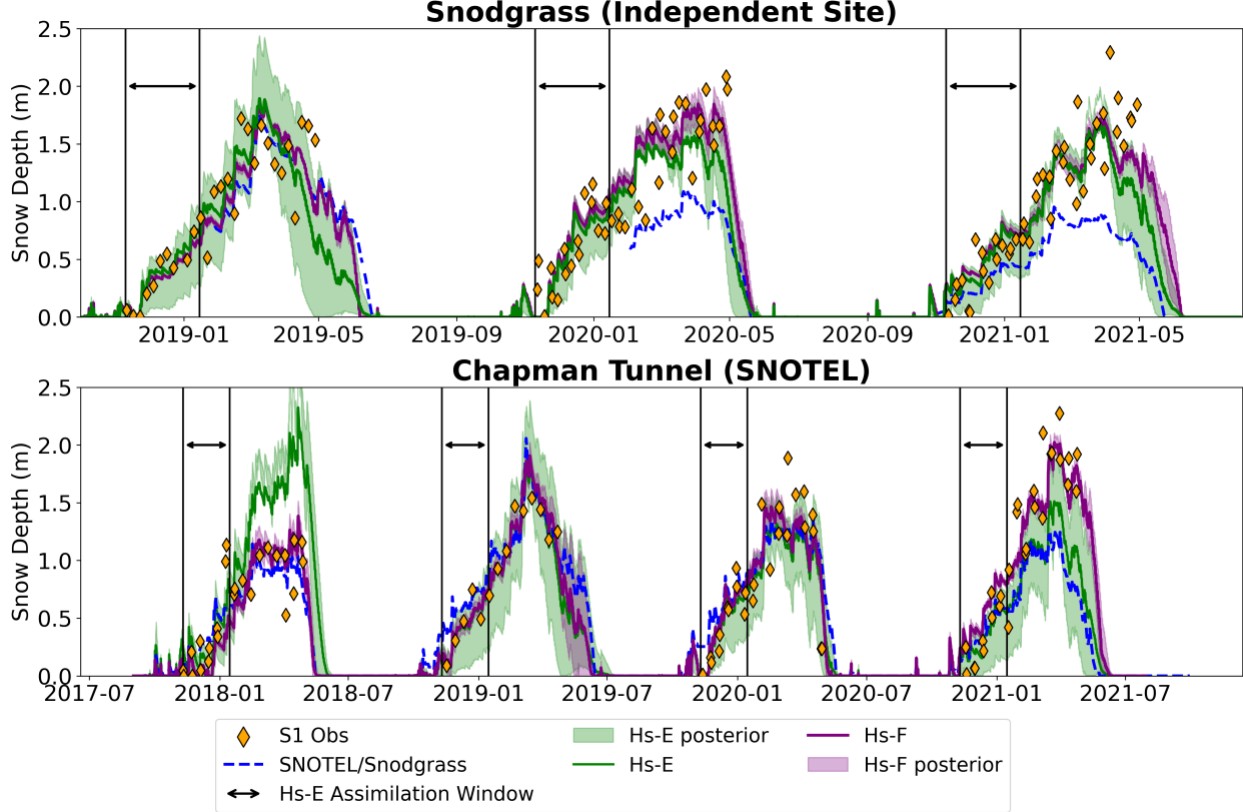

395

**Figure 5:** Temporal analysis results at (top) Snodgrass and (bottom) Chapman Tunnel. Hs-F (purple line) and Hs-E (green line) represent the posterior mean of full and early window experiments, respectively. SNOTEL and Snodgrass station data (blue dashed line) provide snow depth evaluation data. Orange diamonds represent S1 snow depth observations. Shading of respective color indicates one standard deviation of posterior particle spread, and the solid black lines represent the early assimilation window. See Figure S3 for the comparison of these experiments against the model-only run.

The statistical analysis across temporal evaluation sites give insights into the average performance of each DA window experiment. The results show RMSE of 0.29 m for Hs-F and 0.35 m for Hs-E, while $R^2$ is 0.88 for Hs-F and 0.86 for Hs-E (Table 3). Overall, the differences between Hs-F and Hs-E across all sites and evaluation metrics are minimal, suggesting that the accuracy of S1 snow depth retrievals remains consistent regardless of the temporal window. This implies that the retrieval method performs similarly for both the full and early windows.

**Table 3:** Error metrics of posterior mean of Hs-F (full window assimilation) and Hs-E (early window assimilation). Metrics are averaged across all temporal evaluation sites and across four years. See Table S2 for all experiments comparison against the model-only run.

| Metrics | Experiments | |
| --- | --- | --- |
| | Hs-F | Hs-E |
| $R^2$ | 0.88 | 0.86 |
| RMSE (m) | 0.29 | 0.35 |
| Mean Bias (m) | 0.07 | 0.09 |
| MAE | 0.20 | 0.23 |
| RMAD | 0.46 | 0.63 |


While the temporal analysis shows a marginal difference in Hs-F and Hs-E, spatial analysis reveals
a significant disparity in performance, with Hs-F outperforming Hs-E significantly across all
LiDAR flights in most metrics (Table 4). For example, Hs-F demonstrates a lower average RMSE
(0.66 m vs. 0.91 m), lower mean bias (0.05m vs. -0.61m) along with lower MAE (0.42m vs. 0.75m)
and RMAD (0.61 vs. 1.2m) compared to Hs-E. However, Hs-F performs slightly worse in specific
survey dates, such as $R^2$ on 2019-06-10, where Hs-E performs better than Hs-F (0.08 vs. 0.57m).
The temporal and spatial RMSE across all stations and years and LiDAR flights are shown in
Figure 6, with Hs-F showing lower RMSE in all cases except in the 2019-06-10 LiDAR flight.
Across all four LiDAR surveys, the spatial patterns in snow depth are not strongly related (i.e.,
$R^2 < 0.60$) for either Hs-E or Hs-F (Table 4, Figure 7). The density plots (Figure 7c) compare the
posterior mean snow depths from the full-window (Hs-F) and early-window (Hs-E) assimilation
experiments with LiDAR measurements. The density distribution aligns with LiDAR only in
certain cases (Fig. 7c), reflecting both experiments perform consistently poor relative to the LiDAR
snow depths. The overall analysis indicates that S1 has high errors spatially regardless of window,
but the full-window assimilation approach (Hs-F) provides relatively lower errors across spatial
locations and LiDAR dates (Figure 7). Therefore, Hs-F is utilized as the optimal window for the
rest of the data assimilation experiments (see Section 4.3).

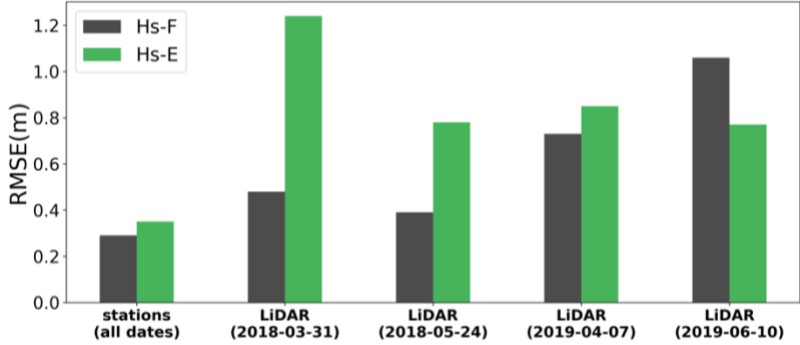


**Figure 6.** Bar chart comparing average RMSE from the Hs-F and Hs-E experiments when evaluated at the 12 snow stations. The average errors are shown for all dates and on each of the four LiDAR flight dates.

**Table 4.** Error metrics for the posterior mean snow depth from the Hs-F and Hs-E experiments when evaluated against snow depth from four LiDAR surveys. See Table S3 for all experiments comparison against the model-only run.

| LiDAR Survey | Exp | $R^2$ | RMSE (m) | Mean Bias (m) | MAE (m) | RMAD (m) |
|---|---|---|---|---|---|---|
| 2018-03-31 | Hs-F | 0.51 | 0.48 | -0.21 | 0.35 | 0.45 |
| | Hs-E | 0.29 | 1.24 | -1.15 | 1.15 | 1.48 |
| 2018-05-24 | Hs-F | 0.33 | 0.39 | 0.08 | 0.21 | 0.88 |
| | Hs-E | 0.37 | 0.78 | -0.59 | 0.69 | 2.55 |
| 2019-04-07 | Hs-F | 0.33 | 0.73 | -0.08 | 0.52 | 0.31 |
| | Hs-E | 0.47 | 0.85 | -0.52 | 0.66 | 0.40 |
| 2019-06-10 | Hs-F | 0.08 | 1.06 | 0.44 | 0.60 | 0.83 |
| | Hs-E | 0.57 | 0.77 | -0.21 | 0.52 | 0.63 |
| **Average** | **Hs-F** | **0.31** | **0.66** | **0.05** | **0.42** | **0.61** |
| | **Hs-E** | **0.42** | **0.91** | **-0.61** | **0.75** | **1.2** |

435

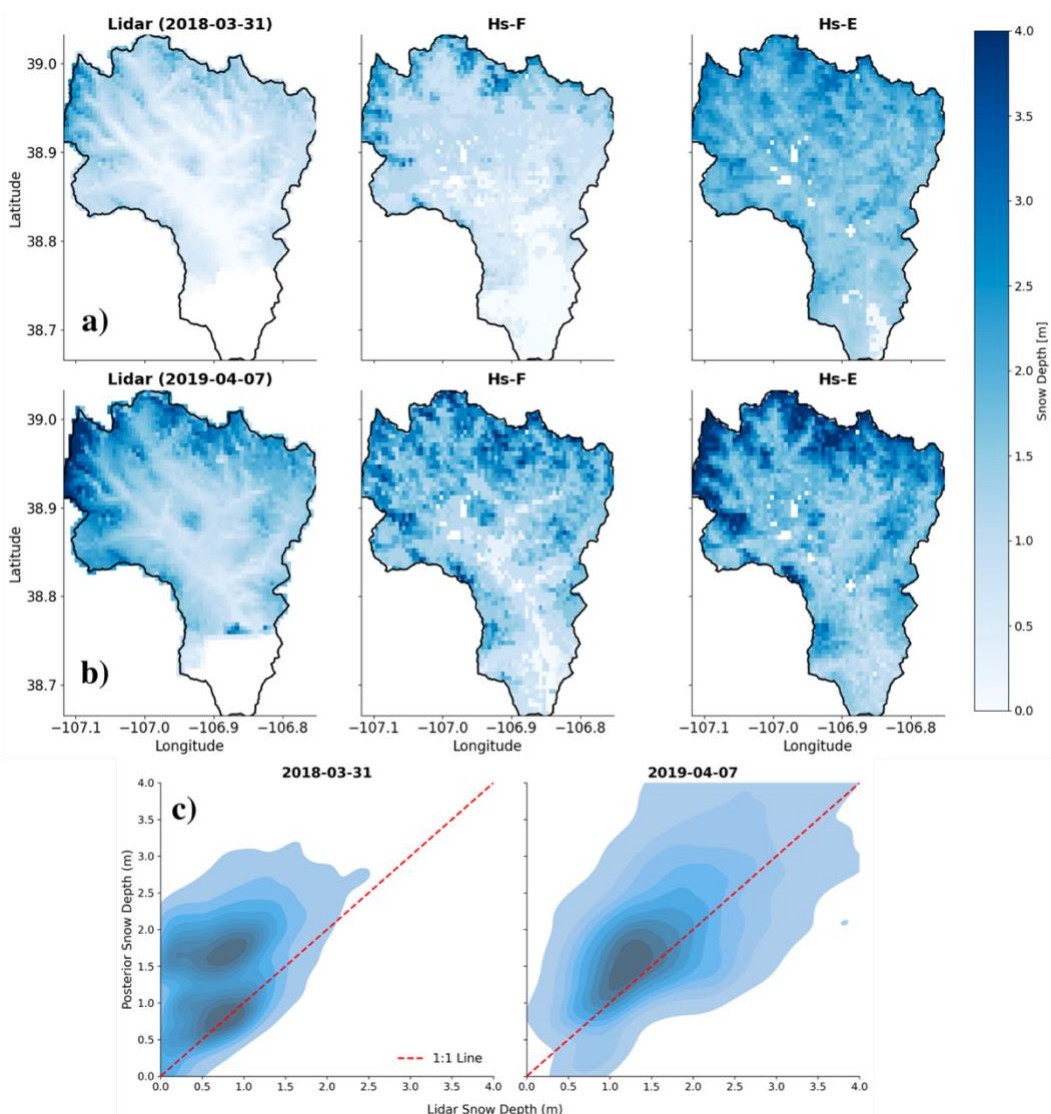

**Figure 7:** The first column in panels a and b shows snow depth from airborne LiDAR near peak snow accumulation. The second and third columns in panels a and b show the posterior mean snow depth from the Hs-F and Hs-E experiments, respectively. Panel C shows the density plot comparing two experiments (Hs-F and Hs-E) against LiDAR measurements. Since both experiments showed similar correlations with LiDAR, they are combined into a single plot. See Figure S5 for the experiments comparison against the model-only run.

## 4.3. Joint Assimilation Experiments:

Finally, we evaluate the joint assimilation of SDD+Hs-F against SDD alone (baseline) to test the utility of S1 snow depth for adding new information to a DA system. We utilized Hs-F instead of Hs-E because errors tended to be lower with Hs-F (see section 4.2). The temporal results of

experiments 3 and 4 of the most representative and independent site are shown in Figure 8. The
results show that the performance of Hs-F (experiment 1), SDD+Hs-F (experiment 3), and SDD
alone (experiment 4) varies across the years. Hs-F performed better than SDD-Hs-F and SDD in
WY 2019 and 2020 and similarly in 2021 for the Snodgrass site. In the SNOTEL representative
site, SDD performed better or similar as Hs-F and SDD+Hs-F in all WYs. The temporal analysis
indicates that joint assimilation and SDD alone experiments perform better than Hs-F minimally
across all metrics (~0.01m). However, the performance of SDD+Hs-F is approximately the same
as the baseline SDD experiment (Table 5), suggesting that joint assimilation of SDD with S1 snow
depth does not significantly enhance performance relative to just assimilating SDD alone.

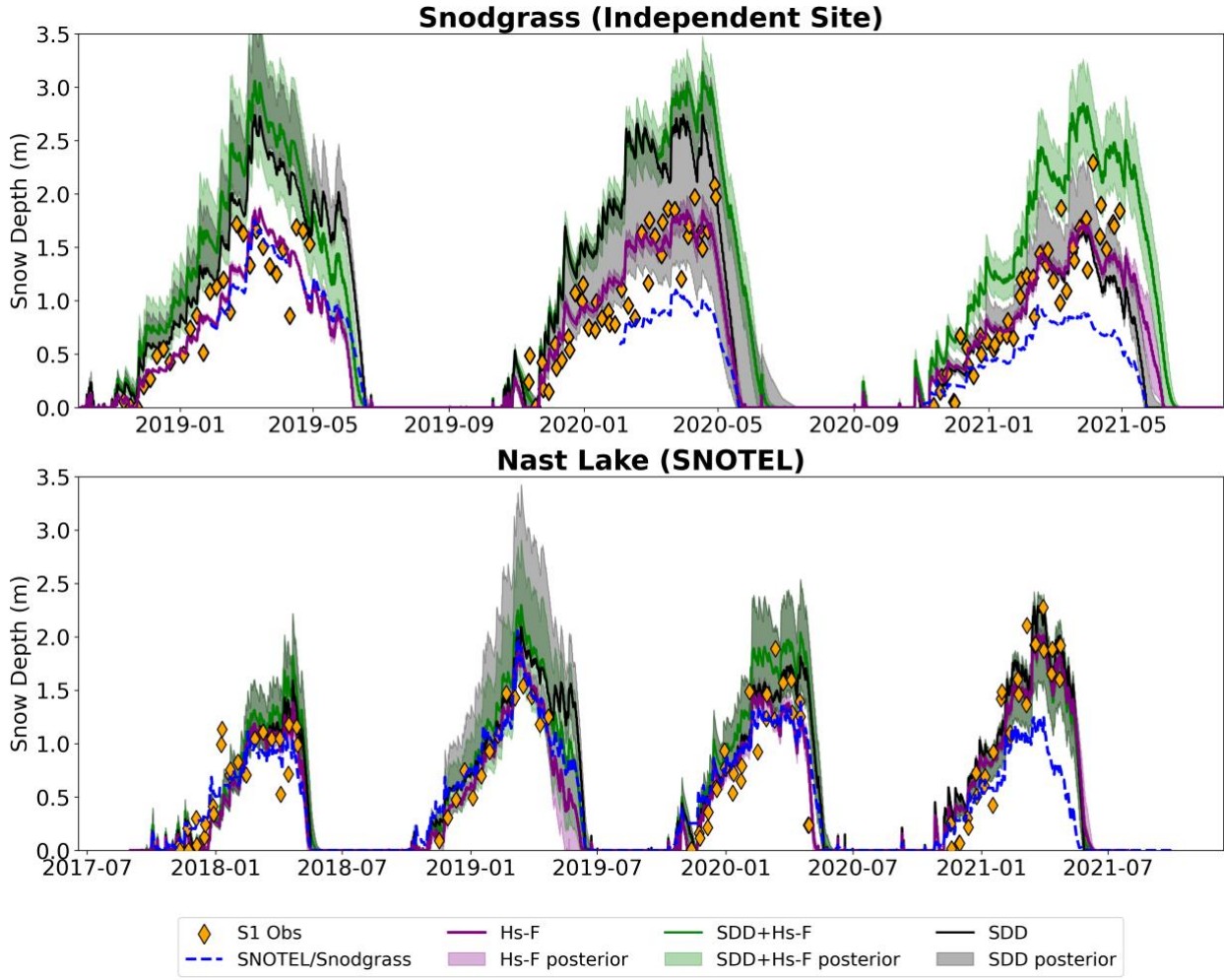


**Figure 8:** Example results of the joint assimilation experiments at (a) Snodgrass and (b) Nast Lake. Hs-F
(purple line) and SDD+Hs-F (green line) represent the posterior mean for each experiment. SDD (black
line) is the posterior snow depth based on assimilating only SDD. SNOTEL and Snodgrass in situ (blue
dashed line) provide snow depth evaluation data. Orange diamonds represent S1 snow depth observations.
**Table 5.** Error metrics of the posterior mean of Hs-F (full window assimilation), SDD+HS-F (joint
assimilation), and SDD. Metrics are averaged across all temporal evaluation sites and across four years. See
Table S2 for all experiments comparison against the model-only run.

| Metrics | Experiments | |
|---------|-------------|---------|
| | SDD+Hs-F | SDD |
| $R^2$ | 0.92 | 0.92 |
| RMSE | 0.28 | 0.28 |
| Mean Bias | 0.15 | 0.14 |
| MAE | 0.19 | 0.19 |
| RMAD | 0.42 | 0.42 |

Spatial analysis provides more insights on univariate (Hs-F or SDD) and joint assimilation
experiments (Hs-F+SDD). Across all metrics, assimilating SDD alone results in higher spatial
performance compared to both Hs-F and SDD+Hs-F. The temporal and spatial RMSE across all
stations, years, and LiDAR flights are shown in Figure 9. The RMSE is 0.41 m for SDD
assimilation averaged across all LiDAR flights and years, which is lower than the RMSE of Hs-F
(0.66 m, Table 4) and SDD+Hs-F (0.70 m, Table 6). Additionally, assimilating SDD alone
demonstrates a stronger relationship with LiDAR observations ($R^2 = 0.72$) compared to Hs-F ($R^2$
$= 0.31$) and SDD+Hs-F ($R^2 = 0.46$), with better spatial pattern alignment across both LiDAR dates
near peak accumulation (Table 6, Fig. 10). Joint assimilation (SDD+Hs-F) shows lower
performance in comparison to Hs-F alone or SDD alone, suggesting that the combination of S1
with SDD does not enhance performance and may potentially degrade it.

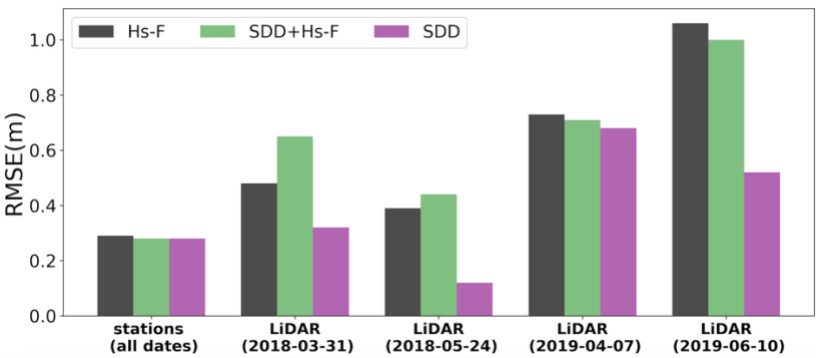


**Figure 9:** Bar chart comparing RMSE for Hs-F, SDD+Hs-F, and SDD experiments of stations
(SNOTEL+Snodgrass) averaged across all stations and all four years, and each LiDAR flight.

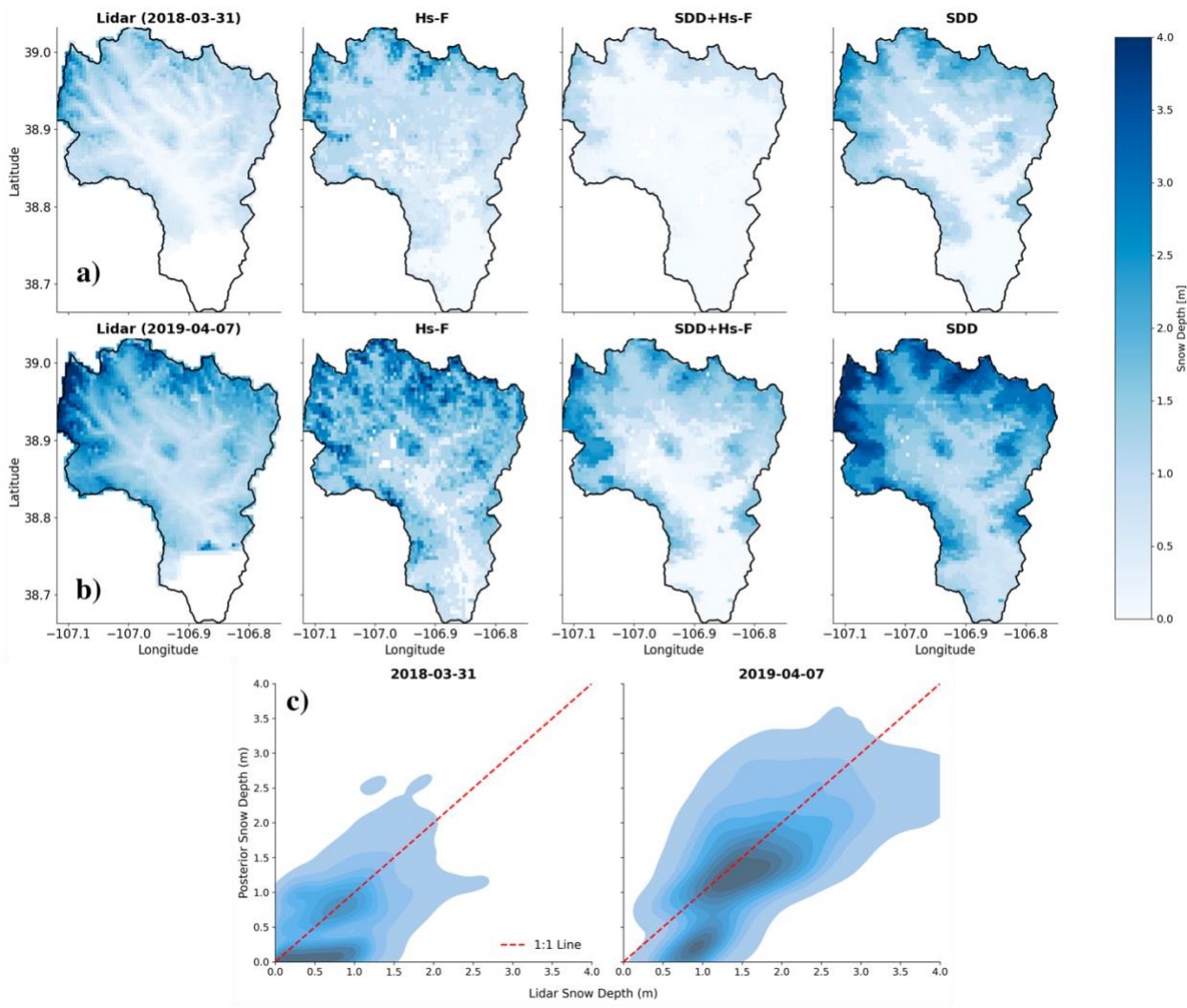


**Figure 10:** The first column in panels a and b shows snow depth from airborne LiDAR near peak snow
accumulation. The second, third, and fourth columns in panels a and b show the posterior mean snow depth
from the Hs-F, SDD+Hs-F, and SDD experiments, respectively. Panel c shows the density plot comparing
three experiments (Hs-F, SDD-Hs-F, and SDD) against LiDAR measurements. Since both experiments
showed similar correlations (or better in the case of SDD) with LiDAR, they are combined into a single
plot. See Figure S5 for the comparison against the model-only run
**Table 6.** Error metrics of posterior mean snow depth from the Hs-F, SDD+Hs-F, and SDD experiments
when evaluated against four LiDAR surveys. See Table S3 for the comparison against the model-only run.

| LiDAR Survey | Exp | $R^2$ | RMSE (m) | Mean Bias (m) | MAE (m) | RMAD (m) |
|---|---|---|---|---|---|---|
| **2018-03-31** | Hs-F | 0.51 | 0.48 | -0.21 | 0.35 | 0.45 |
| | SDD+Hs-F | 0.58 | 0.65 | 0.55 | 0.55 | 0.71 |

| | | | | | | |
|---|---|---|---|---|---|---|
| | SDD | 0.74 | 0.32 | -0.25 | 0.33 | 0.42 |
| **2018-04-24** | Hs-F | 0.33 | 0.39 | 0.08 | 0.21 | 0.88 |
| | SDD+Hs-F | 0.10 | 0.44 | 0.23 | 0.23 | 0.99 |
| | SDD | 0.70 | 0.12 | -0.01 | 0.13 | 0.53 |
| **2019-04-07** | Hs-F | 0.33 | 0.73 | -0.08 | 0.52 | 0.31 |
| | SDD+Hs-F | 0.69 | 0.71 | 0.55 | 0.62 | 0.35 |
| | SDD | 0.70 | 0.68 | -0.48 | 0.54 | 0.32 |
| **2019-06-10** | Hs-F | 0.08 | 1.06 | 0.44 | 0.60 | 0.83 |
| | SDD+Hs-F | 0.49 | 1.0 | 0.71 | 0.71 | 0.85 |
| | SDD | 0.76 | 0.52 | -0.09 | 0.34 | 0.40 |
| **Average** | **Hs-F** | **0.31** | **0.66** | **0.05** | **0.42** | **0.61** |
| | **SDD+Hs-F** | **0.46** | **0.70** | **0.51** | **0.52** | **0.72** |
| | **SDD** | **0.72** | **0.41** | **0.20** | **0.33** | **0.42** |

# 5. Discussion

This study evaluates the utility of S1 SAR-derived snow depth data in the mountainous ERB (Colorado) to support spatiotemporal snow depth mapping within a DA framework. Our analysis highlights inconsistencies in temporal versus spatial errors in S1 snow depth data. While S1 demonstrates lower temporal errors (RMSE = 0.40 m, $R^2$ = 0.74), spatial errors were higher (RMSE = 0.82m, $R^2$ = 0.23) relative to LiDAR flights near or after peak accumulation (Table 2). Temporal errors were lower during the early season but increased over time, particularly during the ablation phase when wet snow conditions become more likely (Figure 4). These discrepancies align with prior studies (Broxton et al., 2024; Hoppinen et al., 2024), which report significant errors in S1 snow depth data under wet snow conditions and poor spatial correlations with airborne LiDAR data (RMSE > 0.7 m, $R^2$ < 0.3). The limitation may also stem from the timing of spatial validation data, as most LiDAR flights are conducted near or after peak snow depth when S1 performance is most likely compromised due to wet snow. However, early-season (Hs-E) experiments show high uncertainty, suggesting uncertainty is likely due to noisy observations without a clear error pattern. Unlike prior studies that proposed excluding flagged wet pixels as a mitigation strategy, we found that removing flagged pixels increased errors, potentially due to the omission of shallow dry snow observations (Hoppinen et al., 2024). Lievens et al. (2022) reported significantly lower errors (~0.25 m) in the European Alps, which is contrary to studies (Broxton

et al., 2024; Hoppinen et al., 2024) focused on the Western US, including our study in the ERB.
The lower errors reported by Lievens et al. (2022) could be due to a higher overpass frequency
and denser validation datasets and needs further investigation. Therefore, while S1 may be useful
in other regions, our findings emphasize its limited reliability in much of the Western US.,
including ERB, and caution against broad generalizations.
Noisy observations are not inherently problematic for data assimilation, provided their uncertainty
is well characterized. Our analysis showed errors increasing with time and did not reveal consistent
spatial or year-to-year error patterns, limiting the development of sophisticated error models
(Figure 4). For simplicity, we treated observational errors as time-invariant for the PBS
assimilation, but we acknowledge that it is possible to include dynamic errors in DA; this is most
straightforward for sequential DA approaches like the particle filter (PF). To assess the potential
benefit of a dynamic error formulation for handling time-varying uncertainty, we compared the
PBS with constant observation error against PF implementations using constant (PFcons) versus
dynamic (PFvar) errors (Figure S1, Table S1). The results showed limited improvement (< 0.040
m MAE). This is consistent with Dunmire et al. (2025), who also tested variable observation errors
when assimilating S1 with an Ensemble Kalman Filter and found that it improved MAE by ~0.025
m at roughly half of their evaluation sites across the European Alps, where Sentinel-1 retrievals
have better performance (Lievens et al., 2022). Given these modest impacts on performance for
sequential DA approaches, we assume our use of a constant observation error did not significantly
impact our results with the PBS, though future work is needed to further confirm this assumption.
We developed our DA experiments based on recommendations by Gascoin et al. (2024), and aimed
to determine the optimal assimilation window (full season vs. early season) with lower uncertainty
for generating spatiotemporal snow depth maps. The comparison between the full-window (Hs-F)
and early-window (Hs-E) approaches showed minimal improvement in error metrics in the
temporal analysis at ground-based stations. Spatial evaluations against LiDAR data resulted in Hs-
F performing better than Hs-E. However, Hs-F increases the likelihood of including wet snow
conditions and higher errors later in the snow season. The early-window approach benefits from
reduced retrieval errors due to its focus on lower snow depth values and assumed higher likelihood
of dry snow in December and January. However, this approach is more effective in regions where
a significant fraction of the total SWE accumulates early in the season (Lundquist et al., 2023). In
contrast, basins like the ERB receive significant snowfall after January, which reduces the early-
season window's ability to predict SWE reliably later in the year (e.g., April-onward). This seasonal
snow accumulation pattern likely explains why the early-window approach is less effective in our
study. This variability underscores the inherent challenges in using early-season conditions to make
broader inferences about snowpack dynamics later in the year. Despite these limitations, the results
show that errors in S1 snow depth retrievals are relatively independent of the assimilation window
in the ERB.
The joint assimilation of S1 snow depth with SDD was conducted to understand what information
is added by S1 and to test whether SDD can provide additional constraints on the DA outcome.
Previous studies successfully used joint assimilation in a DA system with satellite data such as
ICESat-2 (Mazzolini et al., 2024). However, joint assimilation showed limited value with S1 snow
depth data: The errors did not decrease when assimilating S1 depth with SDD. Assimilating SDD
alone showed a higher $R^2$ value and lower errors on average compared to Hs-F and SDD+Hs-F.
There are some caveats and potential limitations in the study. The study did not utilize downscaling
before assimilation; however, we did not see extreme bias in reference runs (model only, Figure
S3-5), which suggests that limited gains from DA are primarily due to S1 observation uncertainty,
rather than a lack of downscaling. Additionally, recent research has shown that assimilation of S1
snow depth can implicitly be used as downscaling (Bachand et al., 2025). The current study
focused on a single basin and a limited number of nearby snow pillow stations over a four-year
period. A more comprehensive evaluation against LiDAR data has already been conducted with
the available LiDAR data in the western U.S. (Broxton et al., 2024; Hoppinen et al., 2024). Similar
to previous studies, we were also limited by the evaluation of LiDAR data since it was only
available at peak snow depth or in the melt season, when S1 has higher errors. This highlights the
importance of expanding the spatial and temporal coverage of evaluation data.
Future research should continue to develop and explore approaches for using Sentinel-1 and other
spaceborne remote sensing platforms for mapping SWE and other related snowpack variables. One
established capability of S1 backscatter data is for wet snow mapping (Cluzet et al., 2024; Gagliano
et al., 2023; Nagler et al., 2016). For snow depth and SWE mapping, the existing Lievens et al.
(2019) algorithm may be improved through corrections with machine learning as proposed by
Broxton et al. (2024). Alternatively, interferometric SAR (InSAR) techniques (Oveisgharan et al.,
2024) have recently been shown to have some potential for mapping SWE with S1 C-band data,
although challenges related to the frequency of satellite passes remain a challenge (Deeb et al.,
2011) and these techniques are likely more reliable with L-band SAR data. The upcoming NISAR
mission will provide L-band SAR data, which may provide new opportunities for accurate snow
depth mapping in mountainous regions that may overcome some of the current limitations of C-
band SAR. However, other limitations may persist for NISAR (e.g., wet snow).

# 6. Conclusion

This study underscores the challenges of using C-Band S1 SAR-derived snow depth data within a
DA framework. While S1 is currently the only high-resolution (<1 km) remotely sensed snow
depth dataset available across the Northern Hemisphere, it demonstrated notable biases and
limitations in snow depth mapping in the East River Basin in the western U.S. These errors were
similar across both full-window and early-window assimilation experiments and align with prior
studies that reported significant spatial biases and retrieval errors. Importantly, we did not observe
a consistent seasonal or interannual pattern in the errors, which limits the development of robust
correction or error models. The joint assimilation of S1 snow depth with SDD data showed limited
or no improvement, suggesting that assimilating SDD alone yields greater accuracy.
While recognizing that our study focused on a single mountain basin, we conclude that the
reliability of the current S1 snow depth retrieval algorithm presents major challenges for a snow
DA system. Future research should prioritize algorithm improvements, explore machine learning
techniques, and conduct additional testing across a wider range of basins and with spatial data.
Enhanced methods for snow depth monitoring would improve understanding of snowpack
dynamics in regions where ground-based observations are sparse or unavailable, supporting better
water resource management and climate impact assessments.

**Author Contributions:**  Data curation and Analysis: BM. Experiment Design: BM, MR, EE.
Writing: The draft was led by BM with key contribution and editing by all co-authors. Funding
Acquisition: MR and ES.
**Code and data availability:** This study used airborne LiDAR snow depth data (ASO, Inc.),
MODIS and ERA5 data (GEE), NRCS SNOTEL, and independent validation data from teams at
the University of Colorado and Oregon State University. The MuSA code, which integrates
FSM, is available at https://github.com/ealonsogzl/MuSA, with FSM originally developed by
Richard Essery. Computational resources were provided by Oregon State University's College of
Engineering.

**Competing interests:** The authors declare that they have no conflict of interest.

**Acknowledgments**
This work was supported by the National Aeronautics and Space Administration (NASA)
Terrestrial Hydrology Program under Award No. 80NSSC22K0685. The authors thank Esteban
Alonso González for making the MuSA code available (https://github.com/ealonsogzl/MuSA) and
Richard Essery for development of FSM. This work utilized resources from Oregon State
University College of Engineering high-performance computing network. The independent
temporal validation data was collected by student teams from the University of Colorado (Eric
Small Hydrology Group) and Oregon State University (CryoSphere Interactions and Geospatial
Hydrology Team). The authors also thank the ASO, Inc. team for providing the airborne LiDAR
snow depth and GEE for facilitating the access of MODIS and meteorological forcing data used
in the study.

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
