# Peer review of "Evaluating the Utility of Sentinel-1 in a Data Assimilation System"

_EGUsphere, 2025_

## Author Comment (AC1)

**Comment on egusphere-2025-978'**, Anonymous Referee #1, 12 Apr 2025

This paper investigates data assimilation (DA) of Sentinel-1 snow depth (S1) and snow disappearance date (SDD) over the East River Basin in Colorado. Overall, the paper is well written and of interest to the science community. The authors find that assimilating S1 data provides limited benefit, whereas SDD assimilation significantly outperforms S1 DA in improving model performance.

We thank the reviewer for taking the time to review our manuscript and for the positive feedback on the writing and relevance of the study. Our responses to the specific comments are provided below.

**Main comments:**

I disagree with the authors broad statement that S1 snow depth (SD) assimilation has "limited utility." Instead, I encourage the authors to clearly explain why the S1 DA does not work well in THIS TEST CASE. These are some main points to support this comment:

1. We know that S1 has limitations in wet snow conditions (known issue), but it can still be valuable for accumulation phases or early-season estimates. The authors say that "basins like the ERB receive significant snowfall after January, which reduces the early-season window's ability to predict SWE reliably later in the year (e.g., April-onward)." So, does this suggest that the East River Basin might not be the best place to test the utility of S1 in general?

**Author's Reply:**

We appreciate the reviewer's feedback and acknowledge that our analysis is limited to a single basin, as previously noted in the discussion section. While Lievens et al. (2019) suggested that Sentinel-1 products perform well across the Northern Hemisphere, there is a growing body of independent studies that are showing lower performance, particularly in the western U.S. (e.g., Hoppinens et al., 2022; Broxton et al., 2024; Ying et al., 2025). Although our study is limited to a single basin, it is consistent with the broader emerging result. In particular, the East River Basin in Colorado serves as a representative and neutral test site for evaluating the performance of Sentinel-1 in snow-covered mountainous regions. This basin presents notable challenges for remote sensing due to its complex topography and occurrences of late-season snowfall. However, it also possesses attributes conducive to Sentinel-1 performance, including moderate forest canopy cover and a cold, dry winter climate that reduces signal attenuation and moisture-related interference. In the revised manuscript, we will clarify that Sentinel-1 performance is regionally dependent and emphasize why the East River Basin represents a valuable case study. While it may be valuable in other basins not assessed here, it is not reliable in much of the Western U.S., including our study area. Therefore, we emphasize the limited utility of this product in much of the Western U.S. and recommend its use with caution when generalizing to the broader Northern Hemisphere.

2. Related to the above point, the current evaluation (comparing S1 DA to ASO lidar data in melt season) is not fair to the "good" (early season) S1 observations, which instead were taken earlier

in the season. Consider: 1) Separating evaluation into accumulation and melt periods; 2) Reporting S1 DA performance specifically during accumulation when S1 is most reliable.

**Author's Reply:**

We are individually reporting errors of each flight date (near peak accumulation and in the melt season); however, we can refer to it clearly in text to make it a fair comparison. Note that we lack "early season" (i.e., before late March) measurements from ASO. Our SNOTEL analysis reveals that a fundamental challenge with assimilating Sentinel-1 snow depth from the early season – since that is the period when snow is more likely to be dry (higher quality SAR measurement) but may be less informative for later season conditions (see Lundquist et al., 2023).

3. Some of the paper conclusions can be associated with the assimilation scheme and not entirely to S1. I think the error analysis (re their question #1) informs that the errors vary over time, is this analysis informing the choice of what measurement error is chosen in the DA scheme? Is the measurement error dynamic (varying with snowpack conditions as suggested by Fig. 4) or constant? The description and choice of the measurement error is of critical importance to indeed evaluate the utility of S1 data in a DA system.

**Author's Reply:**

We appreciate this important observation. The current Particle Batch Smoother implementation uses a constant measurement uncertainty, which does not reflect the time-varying nature of S1 error observed in our analysis. To address this, we tested several uncertainty values ranging from 10cm to 90 cm and used the most representative ones. However, to account for dynamic errors, in the revised manuscript we will test an alternative assimilation approach (e.g., Particle Filter), spatially and temporally, that can incorporate dynamic measurement error. If results show a significant difference, we will revise our approach accordingly. Otherwise, we will include these tests in the supplementary material to clarify whether our conclusions are independent of the choice of assimilation method.

4. In a DA experiment there are always three players: the DA scheme, the model, and the S1. It is not clear how each of them contributes to the results found by the authors. Much responsibility is given to the observations, which could be, but the contribution of DA scheme is not discussed (see for example previous point), nor the models errors/performances are reported. In all tables, authors should - at the bare minimum - inform the readers about the performances of the model. I recommend to always report model (prior to the assimilation) statistics alongside assimilation results. The readers should be able to see how much improvement/degradation is from DA vs. the model skill. I recommend adding model (prior) estimates in all relevant figures (especially Figs 5, 6, 7, 10) and tables (3, 4, 5).

**Author's Reply:**

We appreciate the valuable suggestion and agree that a complete evaluation involves the DA scheme, the model, and the observations. As noted in our response to Comment 3, we plan to test

an alternative DA approach to assess its influence on the results. Regarding the model, we chose not to use it as the primary baseline in this study in the original submission. Instead, we use the DA results with observed Snow Disappearance Date as the reference for performance evaluation. We conducted internal tests comparing the assimilation results to the model alone and found that assimilation performs slightly better. However, to maintain a focused evaluation of Sentinel-1 utility relative to high-accuracy observations, we chose not to include model results in the main text in the original submission. In response to this comment, we will add model (prior) performance in the supplementary material and reference it in the captions and discussion of Figures 5, 6, 7, 10, and Tables 3, 4, and 5.

A few more comments:

- Line 63: Also add the more recent paper by Lievens et al., 2022

**Author's Reply:** We will make this change.

- There is a contradicting reporting of errors in line 90 associated with S1 vs lines 162-165

**Author's Reply:** We thank the reviewer for pointing out this apparent inconsistency. The values reported in Line 90 refer to RMSE from Hoppinen et al. (2024) (~0.92 m) and Lievens et al. (2022) (~0.25 m), while the values in Lines 161–165 refer to mean absolute error (MAE) across varying snow depth ranges, as reported by Lievens et al. (2022). We will revise the text to clarify the difference in error metrics (RMSE vs. MAE) and ensure consistent and non-contradictory reporting to avoid confusion.

- Line 237: "Lower and upper bounds values for precipitation are selected to ensure realistic" how are these bounds implemented? i.e., if a particle ends up being sampled higher than a bound is it set to the upper limit, resampled, or other? Please add explanation

We did not apply any resampling. Instead, we tested multiple sets of bounds for the ensemble generation (i.e., perturbations) and selected values that were wide enough to ensure no observations fell outside the range during assimilation. Since we did not perform any downscaling of precipitation, we chose conservative (higher-than-needed) bounds to account for potential variability and uncertainty. We will revise the manuscript and add number of times the observations are within the bounds.

- Fig. 4: how is the range of errors defined? From the daily values? Please add

**Author's Reply:** The range of errors in Figure 4 represents the monthly mean ± one standard deviation, calculated from daily error values. We will clarify this in the figure caption.

- Figure 5: I assume the gray shading is the spread of the prior particles? What about the posterior? Is it ZERO? Can you add also the posterior spread? My fear is that the chosen means. error (which is critical to know in this paper yet struggle to find what value was used) is likely just too small.

**Author's Reply:** The gray shading in Figure 5 represents the prior particle spread (one standard deviation). The posterior spread is not currently shown but is not zero. To address this, we will revise the figure to include the posterior spread as a shaded region (with a different color) around the posterior mean lines (Hs-F and Hs-E), allowing readers to visualize the reduction in uncertainty after assimilation. We will also clarify the measurement error used in the data assimilation in the methods section and figure caption to ensure transparency.

- Also in Fig 5, 8 legend, "particle" should not be a gray line rather a gray box, and it should also be called "prior particle spread" or something like this.

 **Author's Reply:** Thank you for pointing this out. We will make this change.

- Table 4: How relevant is the correlation metric in this context? Isn't this primarily driven by model and meteo forcings rather than the DA of snowpack early in the season? Similarly, for table 5, why would one observation only (SDD) lead to such a higher value with respect to the values reported in Table 3?? Wouldn't the correlation values be just an artifact of the model temporal variability?

**Author's Reply:** Correlation is used specifically to evaluate spatial patterns in snow depth across the domain on particular aligned dates. We recognize that correlation-based metrics can be influenced by model structure and meteorological forcing hence, to provide a more comprehensive assessment, we also report RMSE, MAE, and other metrics, which directly capture magnitude and bias errors. Regarding the higher $R^2$ values observed with SDD assimilation (Table 5), we interpret this as a result of the strong spatial constraint provided by SDD observations on snow disappearance timing. We will clarify the purpose and interpretation of $R^2$ in the manuscript to avoid confusion

---

## Author Comment (AC3)

**General comment #1:**

In this work the authors devlop an independent validation of the snow depth S1-based products, and study their ability to update a snow model in a basin with high data availability using DA. The paper is generally well written and structured, and is a good contribution to the available literature.

**Author's Reply:**

We thank the reviewer for their thoughtful and constructive comments and for recognizing the contribution of this work to the snow data assimilation literature. Below, we respond to each of the key points raised:

**General comment #2:**

The authors conclude that these snow depth products have a limited ability to update numerical models. Despite this, there does appear to be a signal in the products, albeit a very noisy one. Given the results, I generally agree with the authors, although probably with more sophisticated error models (perhaps with dynamic error models), there could be some potential in this product. The work would benefit from including in the discussion the possibility in the future of improving the quantification of the uncertainty of the observations, a critical point in DA and too often overlooked.

**Author's Reply:**
We understand that more sophisticated treatment of observation error, particularly dynamic error models, might improve the utility of this dataset in data assimilation. To address this, in our existing work, we tested several uncertainty values ranging from 10cm to 90cm and found the uncertainty values that correspond to RMSE of SNOTEL, the most reliable one. However, to account for dynamic error change, we plan to test a dynamic error approach (e.g., Particle Filter), in contrast to the constant error used in our current Particle Batch Smoother implementation. If this approach leads to significantly different results, we will revise our assimilation framework accordingly. If not, we will include the comparison in the supplementary material to demonstrate that our conclusions are independent of the choice of assimilation method.

**General comment #3:**

I have been surprised by the decision to assimilate SDD. I agree that it probably performs similarly to the more standard FSCA. Although I see some problems in areas with ephemeral snowpack, where several "seasons" may occur. Perhaps the reason is to facilitate manipulation of the data by reducing multiple observations to a single observation, but I would like to know if

there is another motivation, and that the discussion reflects this possible source of uncertainty in the ephemeral snowpack areas.

**Author's Reply:**
We appreciate the reviewer's curiosity about our decision to use SDD. Our main motivation was to understand whether Sentinel-1 provides new information beyond what is already provided by commonly available remote sensing (in this case, MODIS SDD). Additionally, we recognize the potential for optical snow cover mapping to complement the Sentinel-1 data (as in Gascoin et al., 2024), which is only available through April 28 and therefore provides no information about snowpack evolution during the melt season. Moreover, Sentinel-1 has known limitations during ablation due to signal degradation in wet snow conditions. Assimilating SDD allows us to incorporate end-of-season information about snowpack disappearance timing, which provides an important constraint on model performance during the melt period. We will revise the discussion to clarify this rationale and also acknowledge the potential limitations of SDD in areas with ephemeral snowpacks, where multiple melt-refreeze cycles may complicate interpretation.

**General comment #4:**

From the DA point of view, the posterior simulations are treated as deterministic ones, while the posterior is a distribution. There are ensemble validation metrics such as the CRPS that are designed to account for the uncertainty of the posterior ensemble. Also, the authors compare the posterior runs among themselves, but it is important to compare with the reference run. Is the error after assimilating S1 equal to that of the reference (not DA), is it even worse? For example, if the error assigned to the observations is high, the prior ensemble will not be constrained at all after analysis (which is not necessarily negative, it would indicate that the observations are noisier than the uncertainty associated with the forcing). These are important questions to be discussed.

**Author's Reply:**
We agree that the posterior ensemble should be treated probabilistically and that metrics such as the Continuous Ranked Probability Score (CRPS) are valuable for assessing the full distribution of the posterior. In the current version of the manuscript, we focused on posterior means for simplicity, but we will expand our validation to include ensemble-aware metrics such as CRPS. This will better reflect the uncertainty in the posterior and improve the interpretation of the assimilation results.

We conducted internal comparisons between assimilation runs and the reference (no DA) run, and found that assimilation generally resulted in slight improvements. To maintain a focused evaluation on the utility of Sentinel-1 relative to high-accuracy observational benchmarks, we

did not include these results in the main text. However, in response to the reviewer's comment, we will include model (prior) performance in the supplementary material and reference it in the captions and discussion of Figures 5, 6, 7, and 10, as well as Tables 3, 4, and 5. This will allow readers to assess the added value (or lack thereof) of assimilation relative to the open-loop model.

**Specific Comments:**

l.15 - Ensemble-based? all models can be run in ensembles. Maybe physically based?

Author's Reply: We will remove ensemble-based and replace it with physically based.

l.47- Maybe include https://doi.org/10.1029/2021WR030271 for a recent example of spaceborne photogrammetry and DA

Author's Reply: We will add recent citations

l.65 - In my opinion, the biggest challenge of S1-based snow depth data is the accuracy of the product itself as proved by the recent independent validations.

Author's Reply: We agree with the reviewer.

l.71 - P.Broxton, remove P

Author's Reply: We will remove P

l.76 - The authors are running 1D DA experiments, I would remove spatiotemporal since it may be confusing. Anycase, S1 DA has been tested before, showing little improvement as in authors work (eg, https://doi.org/10.1029/2023WR035019)

Author's Reply: We agree and will remove "spatiotemporal" for clarity.

l.100 - It shouldn't be a problem to DA S1 data even during melting, if a proper error model is developed.

Author's reply: We will test other error models and present results to understand the performance dependence on the model as detailed in our response to the other reviewer. Specifically, we will conduct a test with varying observational errors as informed by the SNOTEL evaluation.

l.146 - This is the right citation for ERA5 Land https://doi.org/10.24381/cds.e2161bac

Author's reply: We will correct the citation

l.148 - ERA5 land is available since 1950

Author's reply: We will correct the year.

l.151 - It is true that if sufficient information is provided, DA can be used as a downscaling tool. And it's a smart approach to avoid computational cost since one ensemble could potentially be used for many cells (for non-iterative schemes). But I am not convinced that this is the case for S1, according to the results. I would recommend adding something in the discussion about this, since the reference run (simulation without DA) will be very biased due to the complex topography.

Author's Reply: We agree and will add clarification in the discussion. Our results show that Sentinel-1 assimilation does not significantly improve performance compared to the model-only simulation. This suggests that the main limitation is not the lack of downscaling, but rather the reliability of the Sentinel-1 observations themselves. We will highlight this point and emphasize that, in complex terrain, observation uncertainty may dominate over any benefits from downscaling via DA.

l.154 - Is there any reason to choose PBS over other methods? PBS acronym not introduced yet

Author's Reply: We will introduce the acronym in the revised text. Following the approach of Alonso-Gonzalez et al. 2022 (MuSA), we adopted a Particle Batch Smoother over a standard Particle Filter to better leverage sparse and noisy snow observations from Sentinel-1. The batch smoother provides more robust estimates by incorporating information from the entire assimilation window, which is particularly beneficial in complex terrain where sequential filtering may fail due to observation sparsity and particle degeneracy. Additionally, the PBS is well-suited as a downscaling tool (see your previous comment).

l.175 - *… it is well established for guiding model…* reference needed.

Author's reply: We will add a reference, e.g., (Bishay et al., 2023; Guan et al., 2013)

l.227 - FSM2 "more complex" parameterization uses a Monin-Obukhov stability adjustment

Author's Reply: We will add it in the text.

Table 1. What are the parameters of the lognormal distribution?

For the lognormal distribution, we limit the range to 0–5 to keep values realistic and avoid extreme outliers. For example, a quantile of 0.9 yields a value around 4.2, representing a large but still physically plausible uncertainty.

eq.1 - Why the conditional operator Z|Y is repeated (not repeated in the caption)? Similar comment for pv(V).

Author's reply: This is standard notation in **probability density functions (PDFs)**, especially in Bayesian statistics. It should read in the caption that we will edit it.

pY|Z(Y|Z)

- pY|Z: The name of the PDF: "the probability density function of Y given Z"
- (Y|Z): The value where the function is evaluated

So, it reads:

"Evaluate the conditional probability density of **Y** given **Z** at the value Y (conditional on Z)"

l.284 & eq.286 - I am not sure about what you mean here. eq 1 is compatible with multiple observations of different nature. The number of observations of each quantity should not be a problem if the error is properly modeled. Also, should not be the joint likelihood the product (rather than the sum) of the independent likelihoods? Or were you referring to the log-likelihood, where summation (of log terms) is equivalent to multiplying the likelihoods?

Author's Reply: Under the assumption of independence, the joint likelihood should be the product of the individual likelihoods, and in log space, this becomes a sum. Our approach uses the log-likelihood formulation, where we compute the joint likelihood as the average of the log-likelihoods from Hs and SDD. We will clarify this in the method or experiment section.

Section 4.1 - I miss a visual map comparison between ASO and S1, please include it.

Author's reply: We will add visual maps of ASO and S1 comparison on nearby dates.

l.301- Please consider to include a metric that uses the posterior uncertainty, eg Continuous ranked probability score (CRPS)

Author's reply: Thank you for the suggestion. We will compute the additional metrics and consider replacing them with MAE if the CPRS metric provides more meaningful insight. Otherwise, we will include CPRS in the supplementary material for interested readers. To maintain clarity, we aim to limit the number of evaluation metrics presented in the main text.

Table2 - You can not use R2 for validation comparing timeseries (observed vs modeled) of variables that exhibit seasonality like the snowpack. The seasonal pattern forces R to be high (no snow in summer, some snow in winter). This is probably the reason why you are getting high R2 temporally, but low R2 spatially when comparing S1 against lidar. Also, if you're including

summers or long periods without snow, the RMSE, and maybe other metrics, will be of course low. Please clarify/improve the validation strategy.

Author's Reply: We appreciate the reviewer's concern regarding the use of R² for time series of snowpack variables, which indeed can be inflated by seasonal patterns. To avoid this issue, our temporal validation strategy specifically focuses on periods when observed and modeled snow depth data are both available and non-missing. We calculate R², RMSE, MAE, and bias only for dates within the snow season (October 1 to September 30) where overlapping measurements exist. This ensures that metrics are not biased by long periods of snow absence, such as during summer.

Additionally, our spatial validation (e.g., S1 vs. lidar) is computed on individual dates and therefore not subject to seasonal inflation. We will clarify these details in the revised Methods section to better explain the validation strategy and ensure the robustness of the reported metrics.

l.319 - Please provide the value here for the S1 uncertainty estimation

Author's Reply: We will add a value.

Fig5 - In legend, Particle? Is that grey shadow the ensemble standard deviation? maybe call it open loop or prior ensemble? What about the posterior spread? The inclusion of the posterior dispersion of the experiments probably makes the figure too complicated, but there is no mention of posterior uncertainty anywhere in the paper.

Author's Reply: We will call it prior ensembles and add posterior spread. Originally, it was removed to avoid confusion in the figure.

Table3 (and maybe other places as well) - Same comment as for Table2

Author's Reply: Same reply as on Table 2

Table4 - Control experiments? they are not there (despite they should)

Author's Reply: It was a typo and will be removed. Control/no DA runs will be added in the supplement.

Figure7 - Please include the reference run and observations for comparison. c) How are they combined? This is not very standard

Author's Reply : To keep focus on comparing Sentinel-1 with higher accuracy data, we removed model runs, however, we will include model runs in the supplement and refer to them in the main text.

Table5. Consider to add Hs-F even if repeated, its annoying to scroll up and down for comparing

Author's Reply: We will add.

Figure10 c) please review caption *Panel c shows the density plot comparing two experiments (Hs-F, SDD-Hs-F, and SDD)*. Similar comment as for Fig7c.

Author's Reply: We will check and correct accordingly.

l.434 - This is very speculative. If the poor spatial validation metric is because of the timing of the lidar, shouldn't the DA of the early season perform better?

Author's reply: We agree that the statement was speculative; however, our goal was not to definitively attribute the poor spatial validation to the timing of the lidar flight, but to acknowledge that it could be one of several contributing factors. While it is plausible that early-season data assimilation could perform better if aligned more closely with the lidar acquisition, we currently lack sufficient lidar acquisitions at different times to robustly test this hypothesis. We will reword the sentence to reflect this uncertainty more appropriately.

l.449 For DA, it is not a real problem to use noisy observations, as far as you know that they are noisy. I would reformulate this sentence to highlight the importance of developing more sophisticated error models (which involves a proper understanding of the S1 signal, something that should be better investigated).

Author's Reply: In our investigation, we did not observe a consistent year-to-year pattern in the errors, which currently limits our ability to construct sophisticated correction or error models. However, we agree that a better understanding of the Sentinel-1 signal and its uncertainties is essential and needs further investigation. We will add this to our discussion.

l.470 Missing parenthesis

Author's Reply: We will correct it.

l.476 Maybe include https://doi.org/10.5194/tc-18-5753-2024

Author's Reply: We will include the citation.

l.489 According to your results, why are the biases near and after maximum peak SWE? Since DA of Hs-F performs better than Hs-E, someone might argue otherwise.

Author's Reply: Our initial motivation for the experiment was based on our temporal error analysis results that early-season bias is typically lower, and therefore, assimilation using earlyseason snow depth (Hs-E) would lead to lower overall errors. However, our results revealed that even within this seemingly reliable window, Sentinel-1-derived snow depth is not reliable enough to consistently improve SWE estimates. Additionally, in basins like the East River Basin substantial snowfall often occurs after January, limiting the predictive value of early-season observations for peak and late-season SWE (e.g., April onward). We will clarify this rationale in the revised manuscript.

---

## Author Comment (AC4)

**CC1: 'Comment on egusphere-2025-978', Gabriëlle De Lannoy, 11 Apr 2025**

Great research! Just a small note that earlier Sentinel-1 DA work did not assimilate retrievals past February: Girotto et al., 2024 (Sci Tot En); Brangers et al., 2024 (WRR); De Lannoy et al., 2024 (JAMES).

Author's Reply: Thank you very much for your thoughtful comment and for highlighting these important references. We appreciate the clarification regarding the temporal limitations in earlier Sentinel-1 data assimilation studies, including Girotto et al. (2024), Brangers et al. (2024), and De Lannoy et al. (2024). We will revise the manuscript to accurately reflect this and ensure that these key studies are properly cited.

For the ablation period, the empirical Sentinel-1 SD retrievals are not reliable.

Author's Reply: We agree that Sentinel-1 snow depth retrievals are generally less reliable during the ablation period. In our study, we report error metrics separately for accumulation and ablation seasons to reflect these seasonal differences in performance. Despite observational limitations, the data assimilation system can propagate prior information to improve estimates through the melt season with few reliable early-season-to-late-season observations. We will clarify this distinction in the revised text.

---

## Author Response (AR1)

**Response to the Editor's Request**

Manuscript ID: **egusphere-2025-978**

Title: ***Evaluating the Utility of Sentinel-1 in a Data Assimilation System for Estimating Snow Depth in a Mountainous Basin***

Authors: **Bareera Nadeem Mirza et al.**

Dear Editor,

Thank you for your guidance and the opportunity to revise our manuscript. We have carefully addressed all reviewer comments as outlined in our response letters. Some of the major revisions include:

- Testing the particle filter with both constant and dynamic error against the particle batch smoother (PBS). PBS outperformed, and therefore we retained it; results are provided in the Supplement.
- Improving Figures (e.g., Figure 4) and adding a comparison of Lidar with Sentinel-1.
- Enhancing time series figures to display posterior distributions.
- Incorporating an additional certainty-aware metric (CRPSS).
- Justifying the selection of study locations.
- Adding model-only results to the Supplement.
- Correcting minor typographical errors and refining the text for clarity.

We believe these revisions improve the manuscript substantially and hope it is now ready for further evaluation.

With best regards,

**Bareera Nadeem Mirza**

---

## Author Response (AR2)

**Dear Editor,**

Thank you for the comments and the opportunity to revise my manuscript. The reviewer comments and corresponding responses are provided below.

**Reviewer Comment 1:**
Add a note in the abstract clarifying that while S1 may be useful in other regions, it shows limited reliability in much of the Western U.S., including ERB, to avoid overgeneralizing the findings.

**Author Response:**
We agree and have revised the abstract to include a clarifying statement noting that although Sentinel-1 may be valuable in other regions, our results highlight its limited reliability in much of the Western U.S., including the East River Basin, and caution against broad generalizations of DA performance.

**Reviewer Comment 2:**
Clarify the justification for using a static error with the PF. The reviewer suggests explaining why a PF was used instead of a PBS with a dynamic error model, and adding a discussion referencing the preprint https://doi.org/10.5194/egusphere-2025-2306.

**Author Response:**
We appreciate this insightful comment. We have expanded the discussion to clarify our treatment of observation errors and the rationale for including PF tests. Specifically, we now explain that noisy observations are not inherently problematic for data assimilation if their uncertainty is well characterized. Our analysis showed increasing errors over time but no consistent spatial or interannual patterns, which limited the development of sophisticated dynamic error models. For simplicity, we treated observational errors as time-invariant for the PBS but acknowledge that dynamic errors can be incorporated, particularly in sequential DA methods such as the PF. To evaluate their potential benefit, we compared PBS results with PF implementations using constant and dynamic error formulations. The results indicated minimal improvement (<0.040 m MAE), consistent with recent findings (e.g., Dunmire et al., 2025; Lievens et al., 2022). We therefore conclude that using a constant observation error likely did not significantly affect our PBS results, though future studies should further investigate this assumption.

**Sincerely,**
Bareera Mirza